# CAN LARGE LANGUAGE MODELS ASSESS AND REFRAME PSYCHOLOGICAL ATTRIBUTION: A BENCHMARK AND ANALYSIS

## ABSTRACT

According to the reformulated version of the Learned Helplessness theory, an individual who experiences uncontrollable negative events may subsequently develop a negative attributional style, thereby exhibiting greater susceptibility to depressive symptoms. This depressogenic attributional style not only contributes to depressive symptoms but also represents a malleable target for cognitive therapy. Despite its theoretical and practical significance, computational research on attributional cognition remains underexplored due to the lack of large-scale, high-quality datasets and robust evaluation protocols. In this work, we introduce the Attributional Style Transfer Dataset (ASTD) along with dedicated evaluation metrics, the first benchmark designed to model, assess, and reframe attributional explanations at scale. Constructed via a Prevent–Filter–Validate pipeline that integrates LLM-based generation with specialist validation, ASTD contains 42,000 real-world events paired with psychologically grounded attributions spanning seven styles. Using this dataset, we address two key challenges: (1) scalable assessment of attributional style via both supervised classifiers and zero/few-shot LLMs; and (2)attributional reframing and evaluation, where we propose automatic evaluation metrics to quantify psychological validity. Furthermore, we leverage our proposed metrics to construct a preference dataset, fine-tuning LLMs with Direct Preference Optimization (DPO) and achieving substantial gains in reframing quality. Together, our dataset, metrics, and methodology offer a new paradigm for understanding and modeling attributional style, with direct implications for scalable and adaptive mental health interventions.

## 1 INTRODUCTION

The origins and consequences of depression have long been central to psychological research. A key framework for its cognitive underpinnings is Abramson's reformulated learned helplessness model (Abramson et al., 1978), building on Seligman's early work on helplessness (Seligman, 1972). The model explains how repeated exposure to uncontrollable stressors fosters a *depressogenic attributional style*: individuals construe negative events as internal (e.g., "this is my fault"), stable (e.g., "this will always persist"), and global (e.g., "this affects everything"), while attributing positive outcomes to external or transient causes (e.g., "I was just lucky"). Such patterns interact with life stressors—such as illness, discrimination, and daily setbacks—to shape risk, thereby contributing to helplessness and elevating vulnerability to depression, shown in Fig. 1.

Crucially, while a depressogenic attributional style increases risk, it is also a modifiable target in psychotherapy. Cognitive Behavioral Therapy (CBT) is a structured, time-limited psychotherapy that targets the interplay among thoughts, emotions, and behaviors (Abramson et al., 1989; Vassilopoulos et al., 2015). Within CBT, cognitive restructuring can take the form of attributional reframing, shifting maladaptive causal explanations toward less self-critical, more adaptive alternatives while preserving the event's core meaning (e.g., from "This failure proves I'm worthless" to "This was a tough situation and not a reflection of my abilities"). In practice, assessment has relied on traditional instruments—the Attributional Style Questionnaire (ASQ) (Peterson et al., 1982) and the Content Analysis of Verbatim Explanations (CAVE) (Schulman et al., 1989)—which, although foundational, are resource-intensive because they require manual scoring or constrain respondents to fixed-choice

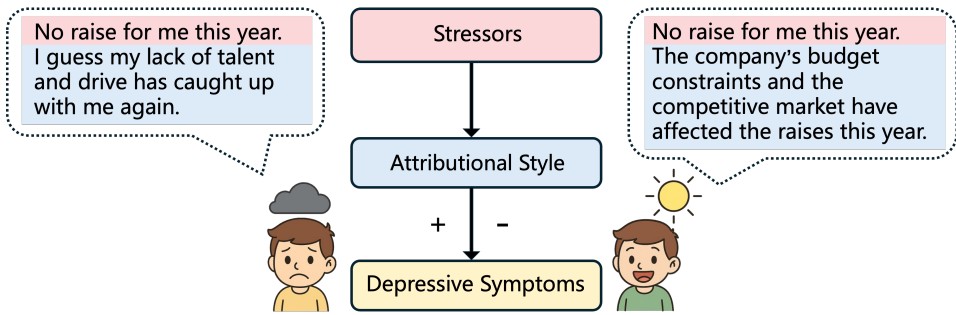

Figure 1: Attributional style modulates how stressors impact depression. For example, attributing no raise to an internal cause (My lack of talent) increases vulnerability, while an external attribution (The company's budget) can act as a buffer.

formats. They also provide limited insight into how attributional styles change over time or respond to intervention. Moreover, unlike conventional text style transfer (Jin et al., 2022), attributional reframing must modify the underlying explanatory logic while preserving the event's meaning and maintaining narrative coherence, making it a challenging and underexplored NLP task and raising open questions about reliable automatic evaluation. Accordingly, two key research questions follow: **RQ1: How can attributional style be efficiently and accurately assessed? RQ2: How can reframing be automatically generated and evaluated at scale?**

Recent LLMs excel at psychological language—interpretation, long-form generation, and naturalistic dialogue—making them suitable scaffolds for high-quality, domain-specific datasets (Demszky et al., 2023; Bang et al., 2023; Goyal et al., 2022). Leveraging this premise, we introduce the Attributional Style Transfer Dataset (ASTD)—the first large-scale dataset focused on attributional style—as a foundation for the challenges outlined above. To minimize hallucination, mitigate bias, and improve diversity beyond fully synthetic data (Li et al., 2024), we adopt an expert-in-the-loop Prevent–Filter–Validate (PFV) paradigm as shown in Fig.3. **Prevent** grounds generation with retrieval-anchored events and semantic constraints. **Filter** applies rule checks and heterogeneous LLM critics to remove inconsistencies, catastrophizing, off-topic content, and duplicates, with expert analyses feeding back as prompt refinements. **Validate** routes low-confidence items to trained experts for majority-vote adjudication. Using PFV, the resulting corpus comprises 42,000 real-world events, each labeled with one of seven attributional styles—internal, external, stable, unstable, global, specific, or neutral—and spanning diverse topics, shown in Fig. 2.

To address RQ1, we compare two ASTD-trained discriminative classifiers and nine LLMs across zero- and few-shot settings. Under our evaluation protocol, the discriminative models achieve the highest average performance for attribution-style classification, whereas LLMs exhibit a clear scaling trend—larger models and few-shot prompting outperform smaller models and zero-shot. These results position ASTD as a rigorous benchmark for evaluating psychologically grounded language understanding in LLMs. For RQ2, we introduce an automatic, four-dimensional metrics—attributional shift, event catastrophizing, coherence, and constructive coping—enabling fine-grained, interpretable, and scalable scoring of reframed outputs. We further derive preference labels from these metrics to fine-tune LLMs via Direct Preference Optimization (Rafailov et al., 2023), yielding consistent gains in reframing quality and psychological validity; expert review and uncertainty analysis corroborate the robustness of both our assessments and the aligned models.

Our contributions are fourfold: (1) Dataset and Pipeline. We release ASTD, the first large-scale corpus of real-world events labeled with seven attributional styles, and introduce a reusable, auditable PFV data-construction pipeline. (2) Benchmark. Using ASTD, we benchmark supervised classifiers and LLMs for automatic style classification; ASTD-trained discriminative models achieve the best average accuracy, and LLMs show clear scaling-law behavior—establishing ASTD as a strong, psychologically grounded benchmark. (3) Reframing Metrics. We propose a four-dimensional, CBT-aligned suite for fine-grained, interpretable, scalable scoring that aligns with expert judgments. (4) Preference Alignment. From metric scores we derive preference labels and fine-tune LLMs via DPO, yielding consistent gains in reframing quality and psychological validity without human annotation.

# 2 ATTRIBUTIONAL STYLE TRANSFER DATASET

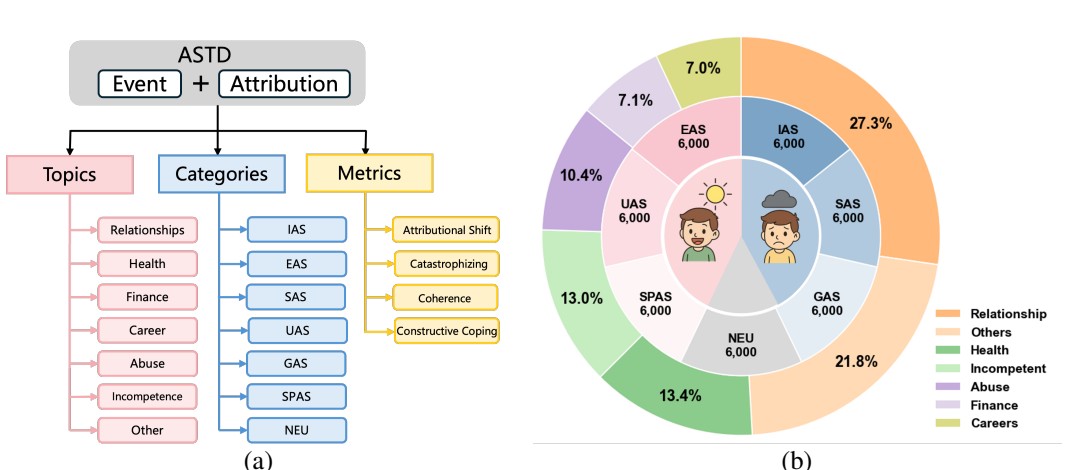

(a)            (b)

Figure 2: (a) Dataset and Metrics Overview. (b) Dataset statistics.

## 2.1 DATA COLLECTION AND ANNOTATION

According to the reformulated Learned Helplessness theory, attributional style is defined along three causal dimensions: locus of control (internal vs. external), stability (stable vs. unstable), and generality (global vs. specific), with detailed definitions provided in Appendix A4. Based on these dimensions, attributions are classified into Internal Attribution Style (IAS), External Attribution Style (EAS), Stable Attribution Style (SAS), Unstable Attribution Style (UAS), Global Attribution Style (GAS), Specific Attribution Style (SPAS), or Neutral (NEU) categories. Individuals who habitually attribute negative events to internal, stable, and global causes (e.g., "it's my fault, it will never change, and it affects everything I do") are particularly susceptible to helplessness and depressive symptoms.

The dataset is organized as Event–Attribution units, where each entry pairs a meaningful event with the subject's causal explanation. JSON-formatted cases are provided in Appendix A3. Events span mental (e.g., "I felt afraid"), social (e.g., "I got a pay raise"), or physical (e.g., "I was in a car accident"), and only events with real impact on the subject are included. The attribution is the subject's stated reason or cause for the event, forming a complete event–cause pair. To construct our dataset, we designed a **Prevent–Filter–Validate** pipeline to emphasize factual grounding, minimize hallucination, and mitigate bias in LLM outputs, as shown in Fig. 3:

**(1) Prevent.** We first employ a retrieval-guided strategy to generate candidate samples. Events are retrieved from multiple publicly available, well-curated real-world datasets (see Appendix A6) and used as factual anchors, reducing the risk of implausible scenarios common in unconstrained generation. These events are then passed to Llama 3.3-70B, a state-of-the-art open-source LLM, which is prompted to produce detailed and logically coherent attributional explanations across seven predefined attributional categories. This approach introduces an explicit

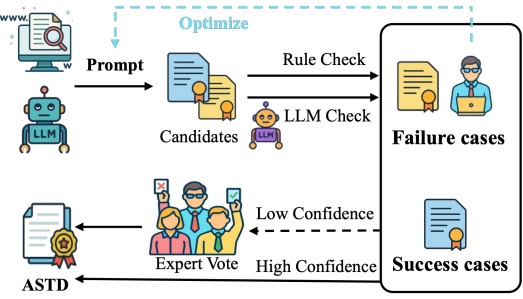

Figure 3: PFV pipeline overview.

retrieval step that substantially reduces hallucination while promoting factual diversity, while the generative component enriches expressiveness and broadens attributional coverage. Thus, our method borrows the key idea of "retrieval–generation coupling" from the Retrieval-augmented Generation (RAG) (Guu et al., 2020; Lewis et al., 2020), yet retains the flexibility and controllability.

**(2) Filter.** Next, we first apply rule-based checks deduplicate events and outputs and enforce first-person perspective, then use the heterogeneous LLM(DeepSeek-R1-32B) to prune inconsistencies, catastrophizing, and off-topic cases; expert analyses of these failures are fed back to improve the prompt for subsequent candidate generation.

**(3) Validation.** Finally, we incorporate human validation for cases where the LLM exhibits uncertainty. Each sample is classified into attributional categories by both Llama 3.3-70B and GPT-4o; we compute the confidence margin between the top-two predicted classes (see Appendix A8). Samples with a margin below 0.2 (See Appendix A8) are flagged as uncertain. Approximately 30% of the dataset falls into this category. These flagged cases are independently reviewed by three trained domain experts—two graduate students in psychology and one graduate student in computer science. All three raters carefully studied the *Guidelines for Rating Transferred Explanatory Style* (see Appendix A12) and completed specialized training before annotation. They conduct independent reviews and reach consensus through majority vote, ensuring reliable and consistent labels while mitigating residual model bias.

## 2.2 DATASET STATISTICS AND ANALYSIS

The dataset comprises 42,000 samples, specifically 12,000 samples for IAS and EAS, 12,000 for GAS and SPAS, 12,000 for SAS and UAS, and 6000 for NEU, as shown in Fig. 2. Our analysis further categorized each sample's trigger event into one of seven types (see Appendix A5), demonstrating comprehensive coverage and enabling downstream applications like therapeutic chatbots for personalized interventions.

To assess the consistency between LLM-generated labels and human judgments, we conducted a labeling agreement study on a randomly sampled subset of 800 examples. Three independent expert raters annotated each sample with one of seven attributional styles. Final human labels were determined via majority vote. We found that LLM-generated labels matched human consensus labels in 94.1% of the cases, yielding a Cohen's $\kappa$ of 0.90—indicating near-perfect agreement and validating the reliability of LLM-based attribution generation.

ASTD exhibits clear advantages over prior resources across both data and task dimensions, see in Table A1. Compared with Cognitive Reframing($\sim$600)(Sharma et al., 2023), ESConv($\sim$1,053)(Liu et al., 2021), PPF($\sim$8.3k)(Ziems et al., 2022b), and PatternReframe($\sim$26.5k)(Maddela et al., 2023), ASTD scales to 42,000 event–attribution pairs and unifies four capabilities: attribution-style classification, reframing generation, strategy rating, and preference-aligned DPO. Grounded in the reformulated learned-helplessness theory, rather than positive-coping heuristics, helping skills, ASTD aligns annotations directly with attributional constructs. This yields richer supervision signals and practical alignment targets, providing a more comprehensive benchmark that simultaneously supports evaluation and intervention-oriented modeling.

## 3 ATTRIBUTIONAL STYLE ASSESSMENT

### 3.1 SUPERVISED METHODS

We fine-tune pretrained BERT (Devlin et al., 2018) and RoBERTa (Liu et al., 2019) on ASTD with a training/test in 80/20 split to preserve a representative held-out distribution. Because evidence for different attribution dimensions can overlap—e.g., "My lack of talent" is simultaneously internal (IAS) and stable (SAS)—we adopt an encoder captures cross-dimension dependencies, while three heads preserve dimension-specific decision boundaries and calibrated posteriors.

Given an event–attribution unit $x$, a shared encoder $h = f_\theta(x) \in \mathbb{R}^d$ feeds three 3-way classifiers indexed by $j \in \{1, 2, 3\}$ for locus, stability, and generality. Each head produces logits $\mathbf{z}_j = W_j h + b_j \in \mathbb{R}^3$ and probabilities $\mathbf{p}_j = \text{softmax}(\mathbf{z}_j)$. With one-hot targets $\mathbf{y}_j \in \{0, 1\}^3$, we minimize the sum of cross-entropies:

$$\mathcal{L}(\theta, \{W_j, b_j\}) = \sum_{j=1}^{3} \text{CE}(\mathbf{y}_j, \mathbf{p}_j) \tag{1}$$

where $\theta$ are encoder parameters and $(W_j, b_j)$ are head parameters.

## 3.2 PROMPT-DRIVEN LLM ASSESSMENT

Our goal is to evaluate LLMs on attributional style classification and to compare them against fine-tuned supervised models, in order to delineate complementary strengths and practical use cases.

We evaluate a range of LLMs on ASTD. For open-source models, we include the Gemma 3 family (Team et al., 2025) (1B/4B/12B), which spans a broad parameter range for examining scaling behavior under a unified architecture, and Llama 3.3–70B (Meta AI, 2024) to cover the large-scale extreme. In addition to non-thinking variants, we also assess reasoning-optimized models, DeepSeek–R1 Guo et al. (2025) (8B/32B). These choices prioritize openness, size diversity, and resource-feasible inference for reproducibility. For an approximate upper bound under current APIs, we further evaluate three proprietary frontier models—GPT-4o, Claude Sonnet 4, and Gemini 2.5-Flash.

Unlike supervised fine-tuning, LLMs are steered via natural-language instructions (Brown et al., 2020). Building on established prompting guidelines (Wang et al., 2022; Ouyang et al., 2022; Chung et al., 2024; Ziems et al., 2024), we adopt a unified prompt template tailored to attributional styles (Appendix A11). We evaluate two settings: (i) zero-shot, where the prompt includes task instructions and style definitions only; and (ii) few-shot (3-shot), where we prepend three fixed, representative event–label exemplars to the same template to probe example-driven inference. Unless otherwise noted, all models share the identical template with a deterministic setting (e.g., temperature $= 0$).

## 4 ATTRIBUTIONAL REFRAMING

Attributional reframing is the core mechanism of the language-guided therapeutic approach known as attributional retraining (Perry, 2017). Rather than denying negative events, it reinterprets them by shifting explanations from internal–stable–global to external–transient–specific (see Appendix A4). This linguistic shift both supports cognitive restructuring and marks the internalization of healthier thinking. Therapist guidance can facilitate early recognition of adaptive interpretations, but durable change depends on the individual's repeated, explicit articulation of revised attributions.

### 4.1 CHALLENGES IN ATTRIBUTIONAL REFRAMING

Attributional reframing is attractive but nontrivial. We highlight two coupled challenges that motivate the next section on evaluation:

**(1) From identification to controlled transformation.** Utterances often embed multiple—even conflicting—attributional cues (explicit and implicit). Effective reframing must first recover the operative stance across locus, stability, and generality, and then shift it in a content-preserving, dimension-consistent manner while remaining contextually and clinically appropriate.

**(2) Open-endedness and lack of a single target.** Unlike classification or retrieval, reframing admits many acceptable outputs for the same input. Standard metrics like BLEU (Post, 2018) or ROUGE (Lin, 2004; Novikova et al., 2017) fail to capture qualities. Robust evaluation therefore requires human judgment or novel metrics tailored to attributional and psychological domain.

### 4.2 EVALUATION METRICS

LLMs have shown promise in handling implicit cues and producing context-sensitive rewrites via prompting or instruction tuning. However, the open-ended nature of attributional reframing—with no single gold rewrite and weak correspondence to surface-overlap metrics—demands an evaluation scheme beyond form-based similarity. We therefore adopt a set of carefully designed, task-aligned metrics grounded in cognitive–behavioral principles. Each reframed sample is scored by an evaluator LLM along four dimensions, each treated as an independent 1–5 classification (1 = poor; 5 = excellent); detailed definitions are provided in Appendix A12.2.

**Attributional Shift** — Degree to which the rewrite shifts from maladaptive to adaptive attributions (internal→external, stable→unstable, global→specific).

**Event Catastrophizing** — Degree to which the rewrite adopts a catastrophizing tone or framing; effective reframes avoid exaggeration and ground the event in realistic, specific context.

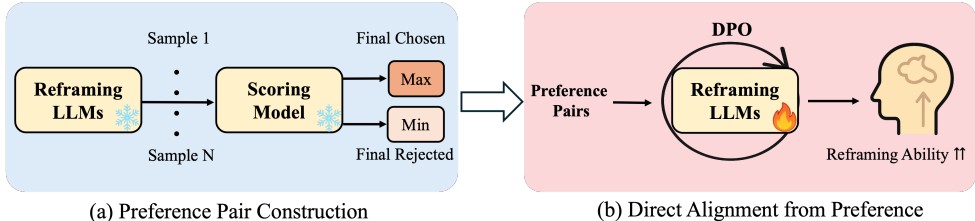

(a) Preference Pair Construction        (b) Direct Alignment from Preference

Figure 4: A frozen LLM generates candidates scored to select best/worst per prompt, then the reframing LLM is DPO-fine-tuned on the resulting preference pairs.

**Coherence** — Degree of logical structure and thematic focus with a clear problem to response flow; all content supports a unified message.

**Constructive Coping** — Presence (explicit or implied) of adaptive coping (e.g., action plan, reappraisal, self-compassion).

We formalize evaluation as aggregating per-dimension scores into a single quality measure per example. Let $s_i^{\text{attr}}, s_i^{\text{cat}}, s_i^{\text{coh}}, s_i^{\text{cope}} \in \{1, 2, 3, 4, 5\}$ denote the four dimension scores in sample $i$. The overall score is

$$s_i = \alpha_{\text{attr}} s_i^{\text{attr}} + \alpha_{\text{cat}} s_i^{\text{cat}} + \alpha_{\text{coh}} s_i^{\text{coh}} + \alpha_{\text{cope}} s_i^{\text{cope}} \qquad (2)$$

with equal weights by default: $\alpha_{\text{attr}} = \alpha_{\text{cat}} = \alpha_{\text{coh}} = \alpha_{\text{cope}} = \frac{1}{4}$.

The unified evaluation structure and an example prompt are provided in Appendix A11. This framework enables a principled evaluation of both the linguistic and psychological adequacy of reframes.

## 4.3 DIRECT PREFERENCE OPTIMIZATION

Direct Alignment from Preferences (DAP) Rafailov et al. (2023) has emerged as a simpler and more stable alternative to Reinforcement Learning from Human Feedback Christiano et al. (2017); Stiennon et al. (2020): instead of learning an explicit reward and running policy optimization, DAP methods update the policy $\pi_\theta$ directly from pairwise preferences. We adopt DPO as a representative DAP method to improve LLM's reframing ability and the full pipeline is summarized in Fig. 4.

**Preference dataset construction.** Given an input prompt $x$, we sample a candidate reframing set $\mathcal{C} = \{y_1, \ldots, y_n\}$ from base LLMs using a fixed sampling strategy (temperature=0.9). Each $y_i \in \mathcal{C}$ is scored by an evaluator LLM described in Func.2; We then select a preferred and a rejected response

$$y^+ = \arg\max_{y_i \in \mathcal{C}} S(i), \qquad y^- = \arg\min_{y_i \in \mathcal{C}} S(i),$$

breaking ties uniformly at random. To avoid weak or noisy pairs, we require a minimum score margin $\Delta S = S(y^+) - S(y^-) \geq \epsilon$ (we use $\epsilon = 0.5$); near-duplicates are filtered by lexical overlap (ROUGE-L $> 0.95$). This yields a set $\mathcal{D} = \{(x, y^+, y^-)\}$ of preference pairs.

**DPO objective.** DPO minimizes a pairwise loss $\ell(x, y^+, y^-, \theta)$ over $(x, y^+, y^-) \sim \mathcal{D}$, updating $\pi_\theta$ directly from preferences without an explicit reward model. The objective function is formulated as:

$$\mathcal{L}_{\text{DPO}}(\theta) = -\log \sigma \left( \beta \log \frac{\pi_\theta(y^+ \mid x)}{\pi_\theta(y^- \mid x)} - \beta \log \frac{\pi_{\theta_0}(y^+ \mid x)}{\pi_{\theta_0}(y^- \mid x)} \right) \qquad (3)$$

where $\sigma$ is the logistic function and $\beta > 0$ controls preference sharpness. Log-probabilities are computed as the sum of token log-likelihoods over the response.

## 5 EXPERIMENTS

Supervised fine-tuning was performed on a single NVIDIA V100 (32 GB). For LLM inference, we used four NVIDIA V100 (32 GB) GPUs with distributed inference. DPO fine-tuning of DeepSeek-R1–8B was conducted on four NVIDIA V100 (32 GB) GPUs using the ms-swift library Zhao et al. (2025). Further hyperparameters of DPO are listed in Appendix A10.

Table 1: Performance comparison across IAS–EAS, SAS–UAS, and GAS–SPAS under 0-shot and 3-shot. We report Acc. $\pm$ Std; top three Acc. are highlighted.

| Model | Param. | Setting | IAS–EAS | SAS–UAS | GAS–SPAS | Avg. |
|---|---|---|---|---|---|---|
| **BERT-base** | 110M | — | 96.89 | 96.89 | 98.22 | 97.33 |
| **RoBERTa-base** | 110M | — | 95.56 | 96.67 | 98.67 | 96.97 |
| **Gemma 3** | 1B | 0-shot | $56.28 \pm 5.09$ | $32.16 \pm 3.01$ | $28.92 \pm 3.32$ | 39.79 |
| | | 3-shot | $43.50 \pm 2.25$ | $31.30 \pm 2.49$ | $25.53 \pm 3.58$ | 33.44 |
| **Gemma 3** | 4B | 0-shot | $45.11 \pm 2.42$ | $44.91 \pm 4.13$ | $42.83 \pm 4.36$ | 44.28 |
| | | 3-shot | $46.10 \pm 3.83$ | $49.67 \pm 5.12$ | $40.84 \pm 4.43$ | 45.54 |
| **DeepSeek-R1** | 8B | 0-shot | $75.32 \pm 2.54$ | $76.98 \pm 3.22$ | $70.44 \pm 2.59$ | 74.24 |
| | | 3-shot | $76.29 \pm 3.35$ | $86.89 \pm 2.84$ | $77.61 \pm 3.59$ | 80.26 |
| **Gemma 3** | 12B | 0-shot | $68.65 \pm 4.27$ | $81.69 \pm 3.42$ | $73.30 \pm 3.52$ | 74.55 |
| | | 3-shot | $73.66 \pm 4.78$ | $86.00 \pm 4.42$ | $80.05 \pm 2.58$ | 79.90 |
| **DeepSeek-R1** | 32B | 0-shot | $83.80 \pm 3.18$ | $82.02 \pm 2.93$ | $77.21 \pm 2.16$ | 81.01 |
| | | 3-shot | $89.74 \pm 1.98$ | $85.92 \pm 2.88$ | $85.94 \pm 2.93$ | 87.20 |
| **Llama 3.3** | 70B | 0-shot | $81.88 \pm 2.60$ | $87.57 \pm 3.21$ | $71.09 \pm 3.33$ | 80.18 |
| | | 3-shot | $91.95 \pm 2.42$ | $96.24 \pm 2.05$ | $88.26 \pm 3.32$ | 92.15 |
| **Gemini 2.5-flash** | — | 0-shot | $71.86 \pm 2.15$ | $80.94 \pm 2.13$ | $84.24 \pm 2.65$ | 79.01 |
| | | 3-shot | $89.64 \pm 2.77$ | $94.20 \pm 1.88$ | $93.64 \pm 2.98$ | 92.49 |
| **Claude Sonnet 4** | — | 0-shot | $83.70 \pm 5.39$ | $93.54 \pm 3.28$ | $87.91 \pm 3.43$ | 88.38 |
| | | 3-shot | $89.96 \pm 2.89$ | $96.33 \pm 2.01$ | $93.92 \pm 2.67$ | 93.40 |
| **GPT-4o** | — | 0-shot | $66.55 \pm 4.09$ | $86.05 \pm 2.62$ | $82.31 \pm 3.53$ | 78.30 |
| | | 3-shot | $92.00 \pm 3.08$ | $96.43 \pm 1.41$ | $92.02 \pm 2.62$ | 93.48 |

## 5.1 Assessment Results and Insight

The results in Table 1 indicate that neither zero- nor few-shot LLMs surpass the fine-tuned RoBERTa and BERT baselines which underscores the utility of ASTD for training compact discriminative assessors. We also observe clear scaling trends and consistent few-shot gains, especially on the hardest GAS–SPAS dimension (e.g., Llama 3.3-70B improves by $\sim 17$ points from 0- to 3-shot). Reasoning-optimized "thinking" models tend to outperform non-thinking variants under the same prompting protocol; for example, DeepSeek-R1-8B matches or exceeds Gemma 3-12B despite having fewer parameters. Frontier APIs GPT-4o, Claude Sonnet 4, Gemini 2.5-Flash cluster around **92–93**.**5** Avg., competitive yet still shy of the supervised baselines.

**RQ1:** When labeled data are available, supervised fine-tuning on ASTD yields the most accurate and efficient attribution assessment with small discriminative models. When labels are scarce, prompt-only LLMs are viable—and improve with scale and few-shot examples. Beyond assessment, ASTD serves as a rigorous benchmark for psychologically grounded language understanding in LLMs.

## 5.2 Empirical Validation of the Reframing Metrics

**Human–LLM agreement.** We assess alignment between human judgments and LLM-based scores using Spearman's rank correlation (appropriate for ordinal scales). For each dimension—Event Catastrophizing, Attributional Shift, Constructive Coping, and Coherence—we sampled a set of 300 reframed pairs spanning a range of quality. Three human raters scored each item on the 1–5 rubric; their mean forms the human reference. Independently, scores were produced by an evaluator LLM, Llama 3.3-70B, under a standardized prompt. We observed strong rank agreement: Catastrophizing ($\rho = 0.946$), Shift ($\rho = 0.927$), Coping ($\rho = 0.982$), and Coherence ($\rho = 0.973$); the mean correlation was $\rho = 0.957$ (Appendix A7).

**Uncertainty and calibration.** To examine the stability of the evaluator, we applied BetaConformal(Qu et al., 2025) to a 200-item test set; each item was scored ten times per dimension by the evaluator LLM(temperature=0). A prediction was deemed *accurate* if the absolute difference between the evaluator's mean and the human reference was $\leq 1/3$ (one-third of a scale point). BetaConformal produced calibrated prediction intervals around the evaluator's mean. We obtain high empirical coverage and a mean interval width of $\approx 4.8\%$ of the 1–5 range, indicating tight, well-calibrated intervals.

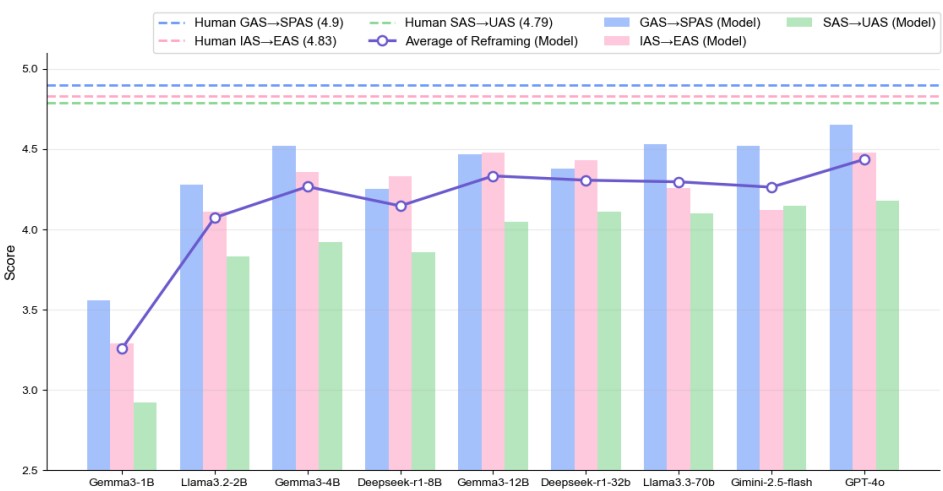

Figure 5: Attributional Reframing performance comparison among LLMs on three domains.

**RQ2:** Strong human–LLM rank agreement together with calibrated, narrow intervals indicates that our automated evaluation is both accurate and consistent, supporting scalable assessment of attributional reframing and reducing reliance on time-intensive human review. These results provide evidence that the proposed metrics can reliably evaluate reframing at scale.

## 5.3 ATTRIBUTIONAL REFRAMING BENCHMARK

As shown in Fig. 5, we evaluate seven open-source LLMs (1B–70B) across Gemma 3, DeepSeek-R1, and Llama reveals clear but non-monotonic scaling: larger models generally reframe better, yet gains plateau at the top end. The reframing axes differ in difficulty, with GAS→SPAS easiest and SAS→UAS hardest. Closed-source models (GPT-4o, Gemini 2.5-Flash) lead overall. However all systems remain below human reference(dashed lines in Fig. 5), indicating substantial room for methods that more reliably execute targeted attributional reframing. These findings motivate a key question: given robust metrics that distinguish high- from low-quality outputs, how can we further enhance LLMs' reframing ability?

## 5.4 DPO RESULTS

Following Sec. 4.3, we fine-tuned DeepSeek-R1-8B and Gemma 3-4B using DPO. Table 2 reports pre-/post-DPO ratings on a 1–5 scale. The improvements are most pronounced for the Attributional Shift metric, which saw a mean increase of $\Delta = 0.565$, indicating that preference alignment effectively fosters more adaptive attributions. Constructive Coping also improved substantially; for example, its score increased by a remarkable +1.113 on the SAS→UAS task. Gains for Catastrophizing were more moderate. In contrast, Coherence scores increased only marginally, a result consistent with the near-ceiling performance of the pre-trained models. The SAS→UAS task—previously identified as the most difficult axis of reframing in our analysis (Fig. 5)—exhibited the largest overall improvement ($\Delta = 0.599$). This suggests that DPO is particularly adept at facilitating complex stability shifts while preserving consistent performance gains in three dimensions.

Table 2: Average ratings before and after DPO fine-tuning on three domains, evaluated across four metrics (higher is better)

| Task | Metric | Pre | Post | $\Delta$ |
|---|---|---|---|---|
| **GAS2SPAS** | Attr. Shift | 4.296 | **4.903** | +0.607 |
| | Catastroph. | 3.807 | **4.156** | +0.349 |
| | Coping | 3.823 | **4.715** | +0.892 |
| | Coherence | 4.964 | **4.990** | +0.026 |
| **IAS2EAS** | Attr. Shift | 4.391 | **4.886** | +0.495 |
| | Catastroph. | 4.104 | **4.343** | +0.239 |
| | Coping | 3.762 | **4.021** | +0.259 |
| | Coherence | 4.964 | **4.994** | +0.030 |
| **SAS2UAS** | Attr. Shift | 3.960 | **4.552** | +0.592 |
| | Catastroph. | 3.904 | **4.380** | +0.476 |
| | Coping | 3.004 | **4.137** | +1.113 |
| | Coherence | 4.708 | **4.921** | +0.213 |

Crucially, these advancements were achieved using preference pairs generated entirely by an LLM-based evaluator with our proposed metrics, requiring no additional human annotation. This result underscores the value and scalability of our evaluation framework and dataset for automated improvement of cognitive reframing in LLMs.

## 6 RELATED WORK

### 6.1 PSYCHOLOGICAL ATTRIBUTION AND DEPRESSION

Depression affects over 280 million people globally and contributes to the loss of 12 billion workdays annually, with an economic burden nearing USD 1 trillion (WHO, 2019; Chisholm et al., 2016). Despite this, more than 75% of individuals with mental disorders in low- and middle-income countries lack adequate treatment access (World Health Organization, 2021), highlighting the urgency of scalable, low-cost interventions. Cognitive Behavioral Therapy remains the most extensively validated treatment for depression, targeting maladaptive thought patterns through structured cognitive restructuring (Beck, 1997; Hollon & Beck, 2013; Dobson & Dozois, 2021). Positive Psychotherapy offers a complementary lens, emphasizing the cultivation of positive emotions and future-oriented optimism rather than solely correcting negative thoughts (Seligman et al., 2006; Nadler et al., 2010; de Jong-Meyer et al., 2007). Attributional theories of depression, particularly the reformulated learned helplessness model (Abramson et al., 1978), propose that individuals prone to depression tend to explain negative events using internal, stable, and global attributions. This maladaptive style is predictive of symptom severity and poorer outcomes (Abramson et al., 1978; Miller & Norman, 1979). Attributional style has since become a key target in both theoretical and clinical frameworks for depression intervention.

### 6.2 ATTRIBUTION STYLE ASSESSMENT

The classical tool for measuring attributional style is the ASQ(Seligman et al., 1979). Respondents attribute a cause for each scenario and rate it on three 1–7 scales—internal vs. external, stable vs. unstable, and global vs. specific. A complementary method, the CAVE (Peterson & Seligman, 1984; Schulman et al., 1989), analyzes naturally occurring speech or writing along the same attributional dimensions. By using archival text instead of self-reports, CAVE reduces confirmation bias, enables studies with populations unable to complete questionnaires, and supports retrospective research. It has revealed links between attributional patterns and outcomes such as academic achievement and health. Despite their value, both ASQ and CAVE require intensive self reports or manual scoring and thus scale poorly (Tang et al., 2013).

### 6.3 DIAGNOSIS OF THOUGHT

Recent cognitively structured prompting methods provide useful context for our work. Diagnosis of Thought (DoT)(Chen et al., 2023) induces cognitive-distortion diagnosis through three staged reasoning steps—separating events from subjective interpretations, generating contrastive causal explanations, and inferring schema-level patterns. The first two stages align conceptually with our attributional assessment and reframing setup, where we explicitly separate events from explanations and elicit causal reinterpretations via reasoning-augmented prompts. Schema analysis, however, targets broader CBT objectives beyond attributional style. HealMe(Xiao et al., 2024) follows a similar multi-stage CBT workflow through simulated client–therapist dialogue. In contrast, our reframing is guided by attributional theory and supports dimension-specific shifts. Thus, while DoT and HealMe model full therapeutic reasoning pipelines, our work focuses on a narrower, theoretically grounded component—attributional style—which can serve as a modular element within larger cognitive reasoning systems.

### 6.4 LLMs FOR PSYCHOLOGY

Large language models (LLMs) such as GPT-4, Gemini, and Llama are built on transformer architectures (Vaswani et al., 2017) and trained on hundreds of billions of internet-sourced utterances (Brown et al., 2020). They have opened up transformative opportunities for psychological science, offering powerful tools for both assessment and intervention. Their ability to generate and interpret human-like language has prompted applications across subfields—ranging from simulating therapy dialogues and detecting belonging concerns (Demszky et al., 2023), to generating growth mindset interventions through prompt-tuning (Handa et al., 2023). However, concerns remain about construct validity, cultural bias, and interpretability (Crum et al., 2013; Bender et al., 2021), calling for the development of domain-specific fine-tuning datasets and evaluation that reflect psychological constructs rather than linguistic fluency alone (Yeager et al., 2022; Binz & Schulz, 2023).

## 7 FUTURE WORK

Future work will extend our framework along three directions.

First, our dataset enables several clinician-supervised, non-intervention downstream applications. Compact discriminative models may support research on attributional assessment tools for identifying potentially maladaptive patterns, while reframing-capable LLMs offer opportunities for exploring clinician-in-the-loop scaffolding systems in controlled, psychoeducational settings.

Second, we will examine how the proposed methodology generalizes beyond attributional style. The PFV pipeline—combining retrieval anchoring, heterogeneous LLM filtering, and uncertainty-driven human validation—is construct-agnostic and can support other cognition-aligned NLP tasks requiring theory-constrained and safety-aware data construction. Likewise, our metric-guided DPO framework provides a reusable alignment strategy for structured cognitive or causal transformations.

Finally, extending ASTD to broader settings remains an important direction. Future work includes constructing variants of the dataset in other languages and cultural contexts, involving more diverse raters, auditing cultural representation, and collaborating with clinical psychologists to identify culturally sensitive or potentially harmful reframing patterns. The modular nature of our framework provides a strong foundation for these extensions.

## 8 CONCLUSION

We introduce ASTD, a 42k-example benchmark built with a Prevent–Filter–Validate pipeline to model, assess, and reframe attributional style at scale. On assessment, compact discriminative models trained on ASTD set the accuracy/efficiency bar, while LLMs show clear scaling and few-shot gains—positioning ASTD as a rigorous, psychologically grounded testbed. On reframing, we propose a four-dimension, CBT-aligned metric suite and verify strong human–LLM agreement, enabling interpretable, automated scoring. Using these metrics to derive preferences, DPO fine-tuning consistently improves reframing quality. Together, ASTD and its evaluation pipeline provide a practical foundation for modeling cognitive theory in language and for informing future research on clinician-guided or therapist-augmented cognitive support systems.

## ETHICS STATEMENT

Our dataset is constructed by referencing several publicly available datasets, with detailed citations provided in the Appendix A6. All referenced datasets permit free use for research purposes. The original data had been de-identified, and during our construction process we additionally applied rule-based filtering to further remove any potential private information as well as violent or explicit content. The dataset is intended solely for evaluating and improving attributional style in order to advance understanding and interventions for psychological well-being. It does not involve human subjects or direct human experimentation.

**Intended Use and Limitations.** This work focuses on modeling attributional style in language and supporting early-stage preventive research. The dataset is not designed or validated for diagnosing depression or delivering therapeutic interventions, and its outputs should not be interpreted as clinical judgments. Any application involving human subjects requires professional oversight and adherence to mental-health safety standards.

**Potential Misuse and Risk Mitigation.** Possible risks include misinterpreting attribution labels as diagnostic signals, deploying reframing models as autonomous therapeutic agents, over-reliance on synthetic reframes in sensitive scenarios, and cultural misalignment when applying models across diverse populations. To mitigate these risks, all resources are provided strictly for research purposes, and any downstream clinical or other high-stakes use requires domain-specific validation and expert supervision.

**Relation to Therapeutic Contexts.** While not a clinical tool, the framework may support future clinician-in-the-loop applications. (i)Discriminative models trained on ASTD can help flag potentially maladaptive attributional patterns, and (ii)reframing-capable LLMs can generate candidate adaptive reframes to expand a clinician's response repertoire. Importantly, clinicians remain fully responsible for interpretation and intervention.

Our work is conducted to benefit human welfare and society, and adheres to the ICLR Code of Ethics.

## REPRODUCIBILITY STATEMENT

To ensure the reproducibility of our work, we provide comprehensive implementation details and experimental specifications throughout the paper and supplementary materials. The experimental setup, including hardware specifications, is described in Section 5 (Experiments), while the data-collection protocols and evaluation metrics are documented in Sections 2.1 and 4.2. In addition, the Appendix includes the full annotator guidebook used during dataset creation. We release our core code in the supplementary materials, and the dataset is anonymously hosted on Hugging Face for public access.

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

## A1 ASTD VERSUS PRIOR COGNITIVE-REFRAMING DATASETS.

Table A1: **ASTD** compared with prior cognitive-reframing or emotional-support datasets.

| Dataset | Data Size | Primary Modeling Tasks | Psychological Basis |
|---|---|---|---|
| Cognitive Reframing of Negative Thoughts (Sharma et al., 2023) | 600 | Language-attribute reframing generation + user-preference study | *Positive coping / CBT principles* |
| ESConv (Liu et al., 2021) | 1,053 | Emotional-support dialogue generation + strategy classification | *Helping Skills Theory* |
| Positive-Psychology Frames (Ziems et al., 2022b) | 8,349 | Positive-reframing generation + strategy tagging | *Positive Psychology* |
| PATTERNREFRAME (Maddela et al., 2023) | 26,500 | Cognitive-distortion classification + reframing generation | *Cognitive Behavioral Therapy (CBT)* |
| **ASTD (Ours)** | **42,000** | **Attribution-style classification + Reframing generation + Strategy rating + Preference-aligned DPO** | *Reformulated Learned-Helplessness Theory* |

## A2 LIMITATION AND FUTURE WORK

Despite its contributions, our work has several limitations. First, ASTD encodes cultural and linguistic norms from its source data and LLMs, potentially limiting generalizability across populations with different attribution styles. Second, while our evaluation framework aligns with human judgments, it relies on LLM-generated ratings that may reinforce biases or overlook edge-case psychological nuances. Thus, future work could create variants of ASTD for other languages and cultures, build stronger theoretical links to psychology, and incorporate human oversight for safe deployment.

## A3 CASE ANALYSIS

```
{
  "utterance": "These headaches are taking a toll on me, and honestly,
    my to-do list is suffocating. The more I struggle to keep up, the
    more I doubt my ability to handle stress. A nagging fear has taken
    holdwhat if this is just my limit, and I'm not cut out for the tough
     times?",
  "event": "Headaches on me",
  "topic": "Health",
  "id": "02996",
  "label": "IAS"
}
```

```
{
  "utterance": "Missing out on celebrating my birthday together was a
    letdown, but this year just seems to have been chaotic for everyone.
     With work and life getting in the way, it's clear people's plates
    are fullit's not about me not being on their minds, just really bad
    timing. Maybe once things settle down, we'll get a chance to catch
    up like we should have. For now, I'm trying to be understanding and
    focus on the fact that they're thinking of me, even if we couldn't
    make it work this time.",
  "event": "Missing out on celebrating my birthday together",
  "topic": "Relationship",
  "id": "00003",
  "label": "EAS"
}
```

## A4 Attributional reframing and Definition of different style.

| Attribution Style | Samples (n) | Word Count Stats | | |
|---|---|---|---|---|
| | | Min | Max | Avg |
| Internal / External | 12000 | 40 | 388 | 80.37 |
| Stable / Unstable | 12000 | 45 | 318 | 84.08 |
| Global / Specific | 12000 | 39 | 458 | 82.63 |
| Neutral | 6000 | 33 | 339 | 83.07 |

**(a)** Attributional reframing.  **(b)** Statistics for posts in each attribution style pair and neutral class.

Figure A1: Overview of (a) attributional reframing direction, and (b) ASTD data distribution.

**Global attributional style (GAS).**  The individual explains the event with a cause that *pervades many life domains*; the cause is taken to affect "all aspects of life," not just the present situation.

**Specific attributional style (SPAS).**  The individual explains the event with a cause that *is limited to this situation or a narrow domain*; the impact does not generalize beyond the particular context.

**Neutral attributional style (NEU).**  The individual *reports the event without giving a cause*; the statement remains factual, avoiding explanations of why it happened or how it affects other areas of life.

**Internal attributional style (IAS).**  The cause is attributed to *the self*—behavioral, physical, or mental characteristics (e.g., personality/traits, behavior/decisions, ability/inability, motivation, knowledge, disability, illness, injury, age, or social/political classifications).

**External attributional style (EAS).**  The cause is attributed to *factors outside the self* (e.g., other people's actions, task difficulty/ease, timing, environment, circumstances, weather, or natural events).

**Stable attributional style (SAS).**  The explanation indicates a *chronic, enduring* cause; given the event, the cause is long-lasting.

**Unstable attributional style (UAS).**  The explanation indicates a *temporary, transient* cause; given the event, the cause is short-lived.

## A5 Categories of Trigger Events and Their Descriptions

Table A2: Categories of trigger events and the corresponding descriptions.

| Category | Description |
|---|---|
| Relationship | Issues related to personal relationships, including conflicts, breakups, or emotional distance between family members, friends, or romantic partners. |
| Health and Medication | Events concerning physical or mental health, such as illnesses, injuries, chronic conditions, and problems related to medical treatment or medication. |
| Financial Stability | Problems involving personal or family finances, such as debt, income instability, or financial crises affecting one's ability to meet basic needs. |
| Jobs and Careers | Difficulties in professional life, including job loss, career stagnation, workplace conflicts, or stress related to career progression. |
| Bias or Abuse | Experiences of discrimination, harassment, or abuse, whether based on race, gender, age, or other factors, often involving power imbalances or unfair treatment. |
| Incompetent | Describes the inability to complete a task or fulfill a role due to a lack of required skills or knowledge. It leads to mistakes, poor outcomes, and frustration for both the individual and others involved. |
| Others | A general category for events that do not fit into the above categories, capturing miscellaneous or unpredictable issues that affect one's well-being. |

## A6   DATASETS USED IN ASTD

Table A3: Datasets used in ASTD.

| Dataset Name | Paper |
|---|---|
| CAMS | (Garg et al., 2022) |
| Tweet Sentiment Extraction | (Maggie et al., 2020) |
| Stress Analysis in Social Media | (Turcan & McKeown, 2019) |
| SALT-NLP | (Ziems et al., 2022a) |
| Reddit Self-Post Classification Task | (Swarbrick Jones, 2020) |

## A7   INTERPRETATION OF SPEARMAN'S CORRELATION COEFFICIENT.

Table A4: Interpretation of Spearman's Correlation Coefficient.

| Spearman's $\rho$ | Interpretation |
|---|---|
| $\geq 0.70$ | Very strong relationship |
| 0.40–0.69 | Strong relationship |
| 0.30–0.39 | Moderate relationship |
| 0.20–0.29 | Weak relationship |
| 0.01–0.19 | No or negligible relationship |

## A8   UNCERTAIN ESTIMATION MODULE

To identify instances where the model exhibits low confidence, we introduce an uncertainty estimation module based on the predicted probability distribution over attributional categories. We formulate the task as a classification problem and query two complementary models, GPT-4o and Llama 3.3-70B, to obtain class probabilities.

**Probability extraction.**   For Llama 3.3-70B, class probabilities are obtained by applying a softmax transformation to the output logits. For GPT-4o, which does not expose token logits, we approximate the distribution through repeated sampling: the model is prompted multiple times with identical input, the predicted category is recorded each time, and the frequency counts are normalized to form an empirical distribution.

**Uncertainty metric.**   Given a probability vector $P = \{p_1, \ldots, p_7\}$, we rank categories by probability and compute the margin between the top two predictions:

$$\Delta = p_{\text{top-1}} - p_{\text{top-2}}.$$

Samples with $\Delta < 0.2$ are deemed *uncertain*, indicating that the model assigns nearly equal probability to multiple categories.

We select 0.2 as a balance between sensitivity and expert workload. A higher threshold (e.g., 0.3) would mark too many cases as uncertain, whereas a lower threshold (e.g., 0.1) would miss genuinely ambiguous examples. Empirically, $\Delta < 0.2$ flags about 30% of the dataset, capturing ambiguous instances while keeping expert review manageable.

**Expert validation.**   All uncertain cases are independently reviewed by three domain experts, who vote on the final attributional category to ensure both label reliability and inter-annotator consistency.

## A9   THE USE OF LARGE LANGUAGE MODELS(LLMS)

During dataset construction, we employed LLMs as key annotation resources within an expert-in-the-loop framework. In our experiments, LLMs also served as evaluators, and our primary research objectives focus on assessing and comparing both open-source and proprietary LLMs. Throughout

manuscript preparation, we used LLM-based tools to check grammar and refine word choice, aiming to improve clarity and maintain scientific rigor. All uses of LLMs were supervised by the authors to ensure accuracy and integrity.

## A10  IMPLEMENT DETAIL

During supervised training, we used a single NVIDIA V100 GPU with 32GB memory. For inference with large language models (LLMs), including `Llama 3.3-70B`, we employed four NVIDIA V100 (32GB) GPUs. The DPO-based reinforcement fine-tuning of `DeepSeek-R1-8B` was also conducted on four NVIDIA V100 (32GB) GPUs, using the `ms-swift` framework.

**Supervised Experiments Configuration**

- **Dataset:** `attribution_multi`, with 7 output classes.

- **Model:** `bert` (alternatives include `roberta` or `best_model`).

- **Training Method:** Standard cross-entropy loss (`ce`) was used, with optional support for `scl` (supervised contrastive loss) and `dualcl` methods.

- **Batch Size:** 16 (training), 64 (testing).

- **Epochs:** 15.

- **Optimizer Settings:** Learning rate of $1 \times 10^{-5}$, weight decay of 0.01.

- **Scheduler:** StepLR with `step_size` = 80 and `gamma` = 0.1.

- **Additional Loss Terms:** Temperature = 0.1, $\alpha = 0.5$ (used in dual-loss setups).

- **Device:** Training was conducted on a single NVIDIA GPU (32GB V100), using CUDA.

**DPO Fine-tuning Configuration**

The training was conducted on four NVIDIA V100 GPUs (32GB each), with the following hyperparameters and settings:

- **Model:** `DeepSeek-R1-Distill-Llama-8B` in FP16 precision.

- **Framework:** `ms-swift`, with LoRA-based parameter-efficient fine-tuning (`lora_rank=8, lora_alpha=32, lora_dropout=0.05`).

- **Dataset:** `Attributional-Style-Dataset`, processed with 4 parallel workers and shuffled.

- **Max sequence length:** 2048 tokens.

- **Batch size:** 1 sample per GPU, with `gradient_accumulation_steps = 4`.

- **Optimizer:** AdamW with $\beta_1 = 0.9$, $\beta_2 = 0.95$, $\epsilon = 1 \times 10^{-8}$, weight decay of 0.1.

- **Learning rate:** 1e-4, with cosine schedule and warmup ratio 0.05.

- **Epochs:** 5.

- **Gradient clipping:** 1.0.

- **Evaluation:** Performed every 100 steps, logging every 5 steps.

- **Checkpointing:** Saved every 100 steps, keeping the latest 2 checkpoints.

- **Precision:** FP16 training with `gradient_checkpointing` enabled.

- **Generation settings:** Temperature = 0.9, Top-$k$ = 50, Top-$p$ = 0.9, Max new tokens = 64.

## A11 Reframing Evaluation Prompt Template and Example

**Prompt Template for Reframing Evaluation**

**1. Dimension Definition**
Coping / Coherence / Attributional Shift / Event Catastrophizing

**2. Rating Scale Definitions (1–5)**
Rate [**DIMENSION NAME**] on a scale from 1 to 5:
**1** = [Definition of a **POOR** reframing on **THIS DIMENSION**]
**2** = [Definition of a **WEAK** reframing on **THIS DIMENSION**]
**3** = [Definition of a **MODERATE** reframing on **THIS DIMENSION**]
**4** = [Definition of a **GOOD** reframing on **THIS DIMENSION**]
**5** = [Definition of an **IDEAL** reframing on **THIS DIMENSION**]

**3. Few-shot Examples (1–5)**
"[Reframed post with **POOR** quality]" → **1**
"[Reframed post with **WEAK** quality]" → **2**
"[Reframed post with **MODERATE** quality]" → **3**
"[Reframed post with **GOOD** quality]" → **4**
"[Reframed post with **IDEAL** quality]" → **5**

**4. Step-by-Step Reasoning Instructions**
STEP 1: **Extract** the aspect **THIS DIMENSION** evaluates (*e.g.*, logic, event).
STEPS 2-4: Conduct logical reasoning steps based on this dimension.
STEP 5: **Output a Single Score** (**1–5**) as a classification task.

Figure A2: Prompt Template for Reframing Evaluation across Dimensions.

---

**COPING_PROMPT**

**Definition:**
**Coping**
Coping measures how well the transferred post shows specific actions or psychological strategies to deal with the described problem. Good coping can be emotional (reframing, hope) or behavioral (taking steps to improve the situation).

**Rating Scale (1–5):**
- **1**: No coping at all. The post describes suffering or frustration but includes no sign of resolution or effort to manage it.
- **2**: Some vague or unclear mention of coping (e.g., "maybe it'll be oka") without direction or action.
- **3**: General emotional reframing or acceptance, but no clear steps.
- **4**: Post contains one or more helpful coping ideas, though they may be a bit vague or lack follow-through.
- **5**: Strong, specific, actionable strategies clearly tied to the problem and showing thoughtful coping.

**Examples:**
- **1** –"I lost my job last week and can't stop thinking about how worthless I am."
- **2** –"My friends have been distant lately. Maybe I just need to let things go and move forward somehow."
- **3** –"I've failed so many times, but I try to remind myself that mistakes are part of growth, even if it's hard to believe right now."
- **4** – "I've been feeling lost lately, so I started journaling and limiting social media to see if that helps."
- **5** – "After losing my job, I made a schedule to update my resume, apply to three jobs per day, and joined a local support group for job seekers."

**Step-by-Step Reasoning:**
1. **Identify the Core Problem** – What situation or emotional struggle is the post about?
2. **Check for Coping Presence** – Does the post show any attempt to manage or deal with the problem?
3. **Determine the Type of Coping**:
   - Emotional? (e.g., reframing, hope, acceptance)
   - Practical? (e.g., seeking help, setting goals, taking specific steps)
4. **Evaluate Specificity and Actionability**:
   - Are the coping strategies vague, general, or specific and concrete?
   - Are they clearly linked to the stated problem?
5. **Assign a Score (1–5)** – Match the coping strategies' clarity and relevance to the rating descriptions above.

**Your Task:**
You are now a strict classifier.
Follow the steps above, then output **only one number (1–5)** that reflects the level of coping strategy in the transferred post. Do not provide explanations.

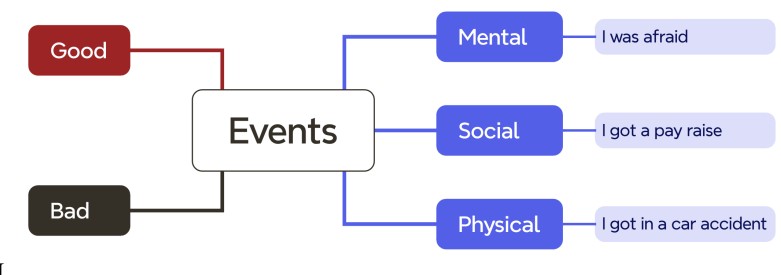

H

Figure A3: Trigger Events

## A12 GUIDELINES FOR RATING TRANSFERED EXPLANATORY STYLE

### A12.1 EVENT AND EXPLANATION

**Trigger Event.** A trigger event is defined as any stimulus that occurs in an individual's environment or within that individual (e.g. thoughts or feelings) that has a good or bad effect from the individual's point of view. Events can be mental (e.g. I was afraid), social (e.g. I got a pay raise) or physical (e.g. I got in a car accident). Events should be unambiguously good or bad from the individual's point of view and may occur in the past, present or hypothetical future. Events that have good and bad elements, neutral events or events that do not affect the Subject should not be extracted. The event must be unambiguously good or bad from the Subject's point of view.

**Explanation.** The explanation refers to the causal statement made by the Subject for the event. Only events which have explicit explanations are to be extracted.
The Subject must express his or her own explanation for that event, and not simply agree with or quote another person's (e.g. therapist or interviewers) explanation. There must be a clear causal relationship between the explanation and the event, and not simply a sequence of events that describe without explaining. The explanation of the event should not be just a proof or justification of the event. The explanation should clearly precede and cause the event.

### A12.2 EVALUATION METRICS FOR REFRAMING

To evaluate the quality of attributional reframing produced by LLMs and assess their alignment with human judgment, we conducted a structured human evaluation. To ensure consistency and inter-rater reliability, we designed a set of standardized scoring rubrics, each corresponding to a specific evaluation criterion.

The following Likert-scale rating schemes (1 to 5) were used:

**Attributional Shift.** This dimension includes three subscales that evaluate changes in causal framing:

- **Internal–External Attribution Rating Scale**
- **Stable–Unstable Attribution Rating Scale**
- **Global–Specific Attribution Rating Scale**

**Event Catastrophizing.** Evaluates how emotionally exaggerated or neutral the reframing is, using the Catastrophizing Rating Scale.

**Coherence.** Measures logical and thematic consistency of the reframed explanation, based on the Coherence Rating Scale.

**Constructive Coping.** Assesses whether the post contains helpful, problem-aligned coping strategies, using the Coping Rating Scale.

Each criterion is rated on a 5-point Likert scale (1 = weakest, 5 = strongest), and is accompanied by concrete examples for each score to guide annotators.

---

**Internal–External Attribution Rating Scale (1–5)**

**Rating Scale:** Rate the locus of attribution on a scale from 1 to 5:

- **1**: Completely internal (e.g., *"I'm lazy," "I'm not good enough," "It's my fault"*)
- **2**: Mostly internal, with a weak or passing nod to external causes
- **3**: Balanced or ambiguous mix of internal and external attribution
- **4**: Mostly external, though some internal reflection remains
- **5**: Fully external — blame or explanation is placed entirely outside the self

**Examples:**

- **Score 1 (Fully internal):**
  *"I failed because I'm just not smart enough."*
- **Score 2 (Mostly internal):**
  *"I wasn't prepared, even though the instructions were a bit confusing."*
- **Score 3 (Mixed):**
  *"I underestimated the time, and the scope changed last minute."*
- **Score 4 (Mainly external):**
  *"The timeline was unrealistic, though I could've spoken up sooner."*
- **Score 5 (Fully external):**
  *"The project failed because the client kept changing requirements and the team was understaffed."*

---

**Stable–Unstable Attribution Rating Scale (1–5)**

**Rating Scale:** Rate the perceived stability of the cause on a scale from 1 to 5:

- **1**: Very stable — the cause is seen as permanent, unchangeable, or part of core identity.
- **2**: Mostly stable — some hint of variation, but mainly framed as enduring.
- **3**: Mixed or ambiguous — hard to tell if the issue is stable or unstable.
- **4**: Mostly unstable — framed as temporary or situation-based, with minor doubt.
- **5**: Clearly unstable — explicitly time-limited, fixable, or unlikely to happen again.

**Examples:**

- **Score 1 (Fully stable):**
  *"I always ruin things. That's just who I am."*
- **Score 2 (Mostly stable):**
  *"I usually mess up public speaking. It's kind of a pattern for me, though maybe I could improve someday."*
- **Score 3 (Mixed):**
  *"I've had good days before, though I still doubt myself a lot."*
- **Score 4 (Mostly unstable):**
  *"This happened because I wasn't mentally present that day."*
- **Score 5 (Strongly unstable):**
  *"I fumbled the interview because I didn't prepare properly this time — I'll adjust my strategy for next week's one."*

## Global–Specific Attribution Rating Scale (1–5)

**Rating Scale:** Rate the degree of generality or specificity in attribution on a scale from 1 to 5:

- **1**: Clearly global — the cause is attributed to the entire self or life (e.g., *"I ruin everything"*).
- **2**: Mostly global — global tone, but with some hint of context or qualification.
- **3**: Mixed — a blend of specific and general attribution; ambiguous boundaries.
- **4**: Mostly specific — attribution is tied to a single domain or situation (e.g., relationships, job).
- **5**: Strongly specific — clearly limited to one context, with no overgeneralization.

**Examples:**

- **Score 1 (Clearly global attribution):**
  *"I always ruin everything I touch, no matter the situation."*
- **Score 2 (Global tone, some qualification):**
  *"I failed this exam, and it just confirms how I'm never really good at anything."*
- **Score 3 (Mixed or ambiguous):**
  *"My friend didn't respond... I probably said something wrong. I always mess up friendships."*
- **Score 4 (Mostly specific):**
  *"The relationship ended because I struggle with vulnerability in partnerships."*
- **Score 5 (Strongly specific):**
  *"I missed that deadline because I overestimated how much I could do in one day."*

## Catastrophizing Rating Scale (1–5)

**Rating Scale:** Rate the degree of emotional catastrophizing on a scale from 1 to 5:

- **1**: The event is described more negatively or catastrophically than before. (e.g., *"destroyed me"*, *"shattered everything"*)
- **2**: Slight increase in emotional emphasis or subjective intensity. (e.g., *"huge disappointment"*, *"major setback"*)
- **3**: Same tone as the original — no significant change in how the event is described.
- **4**: The event is softened somewhat (e.g., less emotionally loaded words), but emotional engagement remains.
- **5**: The event is clearly reframed in a more neutral or rational way — tone is calm, balanced, and reduced in emotional weight.

**Examples:**

- **Score 1 (Strong catastrophizing):**
  *"Getting rejected from the program shattered everything I believed about myself and left me completely hopeless."*
- **Score 2 (Mild exaggeration):**
  *"Missing that deadline feels like a major setback for my future."*
- **Score 3 (No change):**
  *"I forgot some lines during the school performance and felt embarrassed."*
- **Score 4 (Mild softening):**
  *"The professor gave critical feedback on my project, and while it stung, I can see areas for improvement."*
- **Score 5 (Strong reduction of emotional intensity):**
  *"The exam didn't go as planned, despite my preparation. It was a learning experience that highlighted what I need to focus on."*

**Coherence Rating Scale (1–5)**

**Rating Scale:** Rate coherence on a scale from 1 to 5:

- **1** = No coherence at all; completely disjointed, chaotic, or irrelevant.
- **2** = Poor coherence; related theme, but structure is broken and confusing.
- **3** = Moderate coherence; somewhat connected, but with rough or jumpy transitions.
- **4** = Strong coherence; mostly logical and relevant with slight roughness.
- **5** = Excellent coherence; smooth, focused, well-organized progression from problem to reflection/resolution.

**Examples:**

- **Score 1 (Total incoherence):**
  *"I failed my test. The bus was late. Nothing makes sense. I want to paint something green and loud."*
- **Score 2 (Theme present, but logic and flow are badly broken):**
  *"Interviews are hard. I sometimes say the wrong things. My friend likes coffee. Maybe it's just me."*
- **Score 3 (Theme is maintained, but ideas are choppy or inconsistently organized):**
  *"Studying hasn't helped much. Math keeps tripping me up. I feel frustrated. Maybe I'm not a numbers person, but I still want to do better."*
- **Score 4 (Mostly coherent, some roughness or gaps):**
  *"Losing my job was a shock. It's hard not to feel like a failure, especially so early in my career. I'm reminding myself that it's a temporary setback, but it's still tough to stay motivated."*
- **Score 5 (Clear and smooth reflection from problem to resolution):**
  *"Failing my first interview was discouraging, but I know it's a learning experience. I'm reflecting on what went wrong and planning to practice with mock interviews. It's a setback, but not a definition of my worth or future potential."*

**Coping Rating Scale (1–5)**

**Rating Scale:** Rate coping adequacy on a scale from 1 to 5:

- **1**: No coping at all. The post describes suffering or frustration but includes no sign of resolution or effort to manage it.
- **2**: Some vague or unclear mention of coping (e.g., *"maybe it'll be okay"*) without direction or action.
- **3**: General emotional reframing or acceptance, but no clear steps.
- **4**: Post contains one or more helpful coping ideas, though they may be a bit vague or lack follow-through.
- **5**: Strong, specific, actionable strategies clearly tied to the problem and showing thoughtful coping.

**Examples:**

- **Score 1 (No coping):**
  *"I lost my job last week and can't stop thinking about how worthless I am."*
- **Score 2 (Vague coping mention):**
  *"My friends have been distant lately. Maybe I just need to let things go and move forward somehow."*
- **Score 3 (Emotional reframing only):**
  *"I've failed so many times, but I try to remind myself that mistakes are part of growth, even if it's hard to believe right now."*
- **Score 4 (Clear coping, slightly vague):**
  *"I've been feeling lost lately, so I started journaling and limiting social media to see if that helps."*
- **Score 5 (Strong, specific, problem-aligned coping):**
  *"After losing my job, I made a schedule to update my resume, apply to three jobs per day, and joined a local support group for job seekers."*

## A13 Annotator Qualifications

To ensure the reliability of human judgments, our annotation process involved a total of seven trained annotators. Four annotators—two with backgrounds in computer science and two in psychology—performed preliminary labeling following a standardized guideline (Appendix A12). For expert verification(agreement study), we employed three domain specialists: (i) a Ph.D.-level researcher specializing in attributional cognition at the intersection of psychology and machine learning, (ii) a Ph.D.-level researcher working on multimodal large-model emotion analysis, and (iii) a postdoctoral researcher with a doctoral degree in affective psychology from a leading institution, whose expertise spans clinical affective science and affective computing. All experts underwent dedicated training for the rating protocol and performed independent reviews before reaching consensus via majority vote.

