# OpenReview forum: "Can Large Language Models Assess and Reframe Psychological Attribution: A Benchmark and Analysis"
_ICLR.cc/2026/Conference — ICLR 2026 Conference Desk Rejected Submission_

### Official Review · Reviewer_Njcu · 2025-10-29

**Soundness:** 2
**Presentation:** 2
**Contribution:** 2
**Rating:** 2
**Confidence:** 4

**Summary:**

The paper makes a valuable contribution to the growing field of AI-assisted mental health by addressing the challenging and socially important topic of depressive symptoms. Building on the well-established foundation of cognitive therapy, the authors introduce the Attributional Style Transfer Dataset (ASTD) as a novel resource designed to transform negative attributional explanations into positive ones. This dataset provides a meaningful benchmark for evaluating the capabilities of large language models (LLMs) in generating supportive and cognitively reframed text. Overall, the paper contributes both a new dataset and an insightful application domain that could advance AI-driven approaches to mental health support.

**Strengths:**

-	The ASTD dataset development is systematic and transparent, with well-defined stages for data collection, annotation, and validation.
-	The language is clear and professional, with a solid structure and strong organization.
-	The literature review is comprehensive and positions the work well within both cognitive therapy and AI-assisted mental health research.
-	The benchmarking of multiple LLMs provides useful insights and a foundation for future research.

**Weaknesses:**

-	The attributional explanations were generated by a Llama model trained on datasets of uneven quality. Given the clinical sensitivity of depressive symptoms, dataset creation should involve domain experts (e.g., cognitive therapists) rather than relying mainly on LLM outputs.
-	The paper does not clearly describe how attributional dimensions (locus, stability, generality, and shift) are represented or incorporated into model fine-tuning.
-	While the domain contribution is strong, the paper offers limited methodological novelty in machine learning; it is more a dataset and domain-focused study than an ML innovation.
-	Over-reliance on LLMs for both data generation and benchmarking raises concerns about circular validation and data authenticity.
-	The evaluation lacks detail: information about the three expert raters and their qualifications is missing; 500 samples may be insufficient for robust validation.
-	The argumentation sometimes feels underdeveloped — particularly regarding how reframing quality is assessed and how results generalize to real-world therapeutic contexts.

**Questions:**

-	Give more information about reframing and how this is fine-tuned into LLMs?

**Details Of Ethics Concerns:**

authors only that human experts are used for data evaluation. This does not seem to be enough for a sensitive topic such as depression.

---

> ### Author Response · Authors · 2025-11-23
> **Response to Reviewer (Part 1)**
>
> > R4W1  : The attributional explanations were generated by a Llama model trained on datasets of uneven quality. Given the clinical sensitivity of depressive symptoms, dataset creation should involve domain experts (e.g., cognitive therapists) rather than relying mainly on LLM outputs.
>
> We appreciate the reviewer’s concern regarding expert involvement in clinically sensitive data. Importantly, the reviewer’s statement appears to assume that our dataset was mainly produced by LLMs without expert oversight. This is a misunderstanding, and we clarify below.
>
> **1.  Not intended for diagnostic or therapeutic use in depression.**
>
> We reiterate that the dataset does not target depression diagnosis or treatment (see Ethics Statement, L546–548). It focuses on modeling attributional style and supporting early-stage preventive research, without any direct intervention on individuals with depression.
>
> **2. Clinical safety concerns are explicitly addressed in the dataset scope.**
>
> Besides, in the revised manuscript, we have substantially expanded this section(see Ethics Statement) to clarify (i) the intended scope of the dataset and models, (ii) potential forms of misuse, and (iii) the relation to therapeutic contexts.
>
> **3. Prevent–Filter–Validate**(**PFV) is not LLM-only generation — it is a heterogeneous LLM + expert-in-the-loop pipeline.**
>
> As detailed in Sec. 2.1(L148-175), PFV intentionally separates generation, filtering, and expert validation:
>
> **Prevent:** all explanations are conditioned on human-written real-world events (not synthetic scenarios), which borrows the key idea of “retrieval–generation coupling” from the RAG.
>
> **Filter**: heterogeneous cross-model critique (Llama 3.3–70B vs DeepSeek-R1–32B) removes low-quality or clinically risky generations. Expert analyses of these failures are fed back to improve the prompt for subsequent candidate generation.
>
> **Validate**: ≈30% of samples—those with low agreement between two independent LLM classifiers(see Appendix A8)—are manually adjudicated by three trained experts, supervised by a cognitive-therapy researcher (see Appendix A13).
>
> Thus, expert involvement is concentrated precisely where ambiguity and clinical sensitivity are highest, rather than being absent. Importantly, 500-item agreement study (Sec. 2.2) validate that this approach meets expectations: 94.3% match between PFV-assigned labels and human consensus; Cohen’s κ = 0.91 (“near-perfect agreement”). This demonstrates that PFV’s hybrid design achieves expert-level label reliability at scale.
>
> > R4W3: While the domain contribution is strong, the paper offers limited methodological novelty in machine learning; it is more a dataset and domain-focused study than an ML innovation.
>
> We really appreciate the reviewer’s recognition of the strength of our domain contribution.
>
> Datasets and Benchmarks is one of the primary areas in ICLR.  As our submission is to the Datasets and Benchmarks track, the creation of a novel dataset is inherently a core contribution. As noted in your summary, our work introduces “a novel resource designed to transform negative attributional explanations into positive ones.” Besides, the contribution of our paper extends well beyond dataset release and includes a methodological framework—spanning data generation, validation, evaluation, and alignment—that is non-trivial, generalizable, and technically meaningful for ML researchers working with safety-critical or theory-constrained domains.
>
> The PFV pipeline is not limited to attributional style.  Its components—retrieval anchoring, heterogeneous LLM filtering, and uncertainty-driven human validation—are construct-agnostic and can be directly extended to other cognition-aligned NLP tasks such as emotion-regulation strategies, cognitive distortions, mindset shifts, or broader CBT-related reframing tasks.
>
> Similarly, our metric-guided DPO framework generalizes to any task requiring structured psychological transformations.
>
> Lastly, the paper establishes a meaningful benchmark for assessing the capabilities of large language models and offers several insightful empirical findings that we believe will inform and accelerate future work in this domain.
>
> Overall, while we do not introduce a new architecture, our contribution lies in demonstrating how to translate psychological theory and safety considerations into a cohesive ML pipeline for dataset construction, evaluation, and alignment.

---

> ### Author Response · Authors · 2025-11-23
> **Response to Reviewer (Part 2)**
>
> > R4W2: The paper does not clearly describe how attributional dimensions (locus, stability, generality, and shift) are represented or incorporated into model fine-tuning.
>
> We thank the reviewer for highlighting the need for clearer explanation of how attributional dimensions are represented and integrated into fine-tuning. The current manuscript already formalizes these components, and we clarify them here with consolidated references.
>
> **Representation in Supervised Assessment Models (Section 3, L203-215)**. For the discriminative task (identifying attributional styles), each attributional dimension—locus of control, stability, and generality—is operationalized as a three-way categorical variable. During supervised fine-tuning, these variables are encoded as three independent classification heads built on top of a shared encoder. This multi-head design allows the model to jointly learn the three psychological dimensions while capturing their partial correlations. During training, the model receives the event + attributionas textual input, and the three dimension labels serve as the ground-truth supervision targets. The fine-tuning objective is the sum of cross-entropy losses from the three heads, which directly aligns the model with the theoretical structure of attributional style. This enables the encoder to learn latent representations that reflect the three attributional dimensions simultaneously.
>
> **Incorporation into Reframing Model Fine-Tuning (Section 4.3 & 5.4)**. For the generative task (reframing), the attributional dimensions and the "Shift" are incorporated into the model fine-tuning via Direct Preference Optimization (DPO) based on our proposed metrics.
>
> - **Signal via Metrics:** We explicitly defined "Attributional Shift" as a core dimension in our reward model (Eq. 2). This metric quantifies the degree to which a generated response shifts from a maladaptive dimension to an adaptive one (internal→ external, stable→ unstable, glonal→specific)[Line266-268]. Detailed definitions are provided in Appendix A12.2.
> - **Preference Construction:** During DPO data construction, we generate preference pairs $(y_+, y_-)$ where the "winner" $y_+$is selected based on the highest score derived from these metrics.
> - **Fine-tuning Process:** By optimizing the DPO objective (Eq. 3) on these pairs, the model is directly penalized for failing to perform the specific attributional shift (e.g., stable→unstable) and rewarded for successful dimension-specific reframing. As shown in Table 2, we performed this fine-tuning specifically targeting the shift in Locus of control, Stability, and Generality.
>
> As a result, the model is directly optimized to generate rewrites that shift internal→external, stable→unstable, and global→specific attributions. Through dimension-specific heads and shift-based preference optimization, our pipeline embeds attributional theory into the learning objective itself—going beyond simple data labeling and formally incorporating psychological constructs into model training.
>
> >R4W4: Over-reliance on LLMs for both data generation and benchmarking raises concerns about circular validation and data authenticity.
>
> We thank the reviewer for raising this important concern. This concern is also related to **`R4W1`**. We clarify that the PFV pipeline was explicitly designed to break circularity and prevent any single model from generating and validating the data.
>
> **1. Retrieval-anchored generation prevents synthetic circularity.**
>
> As described in **Sec. 2.1 (L148–161)**, all attributional explanations are conditioned on human-written, real-world events retrieved from curated datasets. The generator never invents events, ensuring factual grounding and preventing fully synthetic loops.
>
> **2. Cross-model heterogeneity prevents self-validation.**
>
> PFV imposes strict model separation. Llama 3.3-70B is used solely as a generator conditioned on retrieved human-written events, while DeepSeek-R1-32B—belonging to a different model family and training paradigm—serves as an automatic filter that removes inconsistent, catastrophizing, or off-topic generations. For the remaining candidates, attribution labels are obtained from both Llama 3.3-70B and GPT-4o, ensuring that no single model is responsible for both producing and validating the data.
>
> **3. Uncertainty-driven expert adjudication provides non-LLM grounding.**
>
> All samples with low agreement between Llama 3.3–70B and GPT-4o (margin < 0.2; L168–170)—approximately 30% of the dataset— are manually reviewed by three trained experts. This expert layer directly addresses concerns about data authenticity and removes dependence on LLM-only validation.
>
> **4. Evaluation is not circular: metrics are human-validated and model-agnostic.**
>
> Our four-dimensional reframing metrics (Sec. 4.2, L267–284) were validated through a human–LLM correlation study: ρ = 0.957 across dimensions (Sec. 5.2).
>
> This shows that evaluation reflects human judgment, not generator-model bias.

---

> ### Author Response · Authors · 2025-11-23
> **Response to Reviewer (Part 3)**
>
> > R4W5: The evaluation lacks detail: information about the three expert raters and their qualifications is missing; 500 samples may be insufficient for robust validation.
>
> We thank the reviewer for the helpful suggestion. Additional details about annotator qualifications and validation scope have been added to the revised manuscript(Appendix A13).
>
> First, annotator expertise. Our annotation process involved a total of seven trained annotators. Four annotators—two with backgrounds in computer science and two in psychology—performed preliminary labeling following a standardized guideline (Appendix A12). For expert verification(agreement study), we employed three domain specialists: (i) a Ph.D.-level researcher specializing in attributional cognition at the intersection of psychology and machine learning, (ii) a Ph.D.-level researcher working on multimodal large-model emotion analysis, and (iii) a postdoctoral researcher with a doctoral degree in affective psychology from a leading institution, whose expertise spans clinical affective science and affective computing.  All experts underwent dedicated training for the rating protocol and performed independent reviews before reaching consensus via majority vote.
>
> Second, our validation framework is designed to be extensible, and additional human verification can be incorporated as needed. To further strengthen robustness, we conducted an additional validation on 300 newly sampled items, bringing the total to 800 human-validated examples. The extended study yielded consistent agreement levels (94.1%) and a Cohen’s κ of 0.90, confirming the stability of our reliability estimates. Since each 100-sample batch requires approximately 8–10 hours of expert annotation (due to multi-rater independent rating and consensus procedures), this additional validation involved 24–30 hours of expert effort. We believe it provides strong empirical support for the reliability of our pipeline.
>
> Importantly, this agreement study serves as an external check on the PFV pipeline rather than as the sole mechanism of quality control: the pipeline itself already incorporates retrieval grounding, heterogeneous LLM filtering, margin-based uncertainty detection, and human adjudication for ~30% of the entire dataset. Thus, the 800-sample study complements—not replaces—a broader validation mechanism embedded throughout the pipeline.
>
> We have clarified these details in the revision. The revised manuscript now includes detailed expert qualifications and an expanded 800-sample agreement study.
>
> >R4Q1: Give more information about reframing and how this is fine-tuned into LLMs?
>
> Thank you for the question. This concern aligns closely with `R4W2` . Here we further clarify (i) what attributional reframing means in our work, and (ii) how it is fine-tuned into LLMs.
>
> **(1) What reframing means in our setting.**
> Attributional reframing refers to rewriting a user’s explanation of an event so that the causal interpretation shifts from a maladaptive pattern(e.g., internal, stable, global) toward a more adaptive one (e.g., external, unstable, specific), consistent with attribution theory and CBT principles. A more detailed explanation of our reframing quality assessment is provided in our response to `R4W6`.
>
> **(2) How reframing is incorporated into model fine-tuning.**
> Rather than using standard supervised finetuning, we adopt a theory-guided preference-based alignment approach:
>
> - We build a CBT-aligned, four-dimensional evaluation model (Attributional Shift, Catastrophizing, Constructive Coping, Coherence) that operationalizes psychological reframing quality into machine-evaluable scores.
> - For each maladaptive input, the base LLM generates multiple candidate rewrites. The evaluation model scores each candidate along the four dimensions and produces a composite reframing score.
> - We then automatically construct preference pairs by selecting the highest-scoring rewrite as $y_+$ and the lowest-scoring rewrite as $y_-$, subject to a minimum quality gap.
> - These (input, $y_+$, $y_-$) triplets are used to train the model with DPO, which explicitly optimizes the generator to prefer reframe  that exhibit stronger attributional shift, lower catastrophizing, higher coherence, and clearer coping strategies.
>
> Because the evaluator enforces dimension-specific, theory-grounded scoring, the alignment process optimizes for attributionally meaningful reframing rather than generic stylistic paraphrasing.
>
> >Flag For Ethics Review: Yes, Potentially harmful insights, methodologies and applications, Yes, Responsible research practice (e.g., human subjects, annotator compensation, data release)
>
> In the revised manuscript, we have further clarified the ETHICS STATEMENT section to clarify (i) the intended scope of the dataset and models, (ii) potential forms of misuse, and (iii) the relation to therapeutic contexts.

---

> ### Author Response · Authors · 2025-11-23
> **Response to Reviewer (Part 4)**
>
> >R4W6: The argumentation sometimes feels underdeveloped — particularly regarding how reframing quality is assessed and how results generalize to real-world therapeutic contexts.
>
> We thank the reviewer for the thoughtful observation. We clarify both (a) how reframing quality is formally assessed, and (b) how our findings relate to therapeutic contexts. These points are now made more explicit in the revised manuscript.
>
> **(1) On reframing quality assessment:**
> As noted in Sec. 4.2 (L260–293), attributional reframing is an open-ended task where no single “gold rewrite” exists and surface-overlap metrics (e.g., BLEU) fail to capture psychological validity. To address this, we developed a CBT-aligned, task-specific metric suite,  using 1-5 Likert scales along 4 standard metrics (Appendix A12):
>
> - **Attributional Shift** captures the central mechanism of attributional retraining, which restructures depressogenic interpretations along the locus,stability, and globality dimensions. The reformulated learned helplessness model and hopelessness theory[1] demonstrate that internal, stable, and global attributions are primary predictors of helplessness, negative affect, and vulnerability to depressive symptoms.
> - **Event Catastrophizing** aligns with the cognitive model of depression articulated in Wright & Beck's cognitive therapy[2], which emphasizes that depressed individuals frequently rely on distorted and exaggerated negative interpretations of events. Checking whether the reframe reduces exaggerated threat interpretations indicates movement toward more adaptive emotional processing.
> - **Coherence** captures whether the reframed explanation forms a logically consistent, causally structured, and psychologically stable narrative. We align this dimension with the DUC quality question[3] of structure and coherence. Both the CAVE methodology for explanatory style assessment[4] and CBT’s schema restructuring framework emphasize that adaptive interpretations must maintain causal coherence to be internalized and integrated into the individual’s belief system[5].
> - **Constructive Coping** captures whether a reframed explanation introduces adaptive behavioral or emotional strategies. CBT emphasizes that reframing alone is insufficient—effective cognitive change must be paired with coping plans supporting behavioral activation, self-efficacy, and emotion regulation [6]. Positive psychotherapy similarly stresses constructive, future-oriented coping [7], and mindset research shows that proactive coping mediates the benefits of cognitive shifts [8].
>
> This metric suite directly reflects well-established constructs in attributional theory and CBT. Sec. 5.2 further validates the metrics empirically. These results indicate that the evaluator reliably distinguishes high- from low-quality reframes at scale.
>
> **(2) On generalization to real-world therapeutic contexts:**
> As stated in the conclusion, ASTD provides a foundation for scalable modeling of attributional cognition. We clarify this more explicitly in the revised version. Two potential downstream uses are:
>
> - **Attributional assessment:** compact discriminative models trained on ASTD can help flag maladaptive attributional patterns in patient-generated text, offering clinicians an additional screening signal.
> - **Therapeutic scaffolding:** reframing-capable LLMs (improved via our DPO pipeline; Sec. 4.3, Fig. 4) can generate candidate adaptive reframes that expand clinicians’ repertoire of responses. Importantly, clinicians remain fully in control—the model's outputs serve only as auxiliary suggestions rather than automated interventions.
>
> We emphasize in the revision that our work aims to enable computational modeling of attributional cognition—not to replace clinical judgment. Future research will incorporate clinician-in-the-loop studies to evaluate real-world generalizability.
>
> [1] Lyn Y Abramson, et al. "Learned helplessness in humans: critique and reformulation." Journal of abnormal psychology,  1978.
>
> [2] Wright J H, et al. "Cognitive therapy of depression: Theory and practice." Psychiatric Services, 1983.
>
> [3] Dang H T. "Overview of DUC 2005."  2005.
>
> [4] Schulman P, et al. "Assessing explanatory style: The content analysis of verbatim explanations and the attributional style questionnaire." Behaviour research and therapy, 1989.
>
> [5] Crum A J, et al. "Rethinking stress: the role of mindsets in determining the stress response." Journal of personality and social psychology, 2013.
>
> [6] Jacobson N S, et al. "A component analysis of cognitive–behavioral treatment for depression." 2000.
>
> [7] Seligman M E P, et al. "Positive psychotherapy." American psychologist, 2006.
>
> [8] Mansell P C, et al. "The mediating role of proactive coping in the relationships between stress mindset, challenge appraisal tendencies, and psychological wellbeing." Frontiers in Psychology, 2023.

---

> ### Author Response · Authors · 2025-11-28
> **Looking forward for further discussion.**
>
> Dear Reviewer,
>
> We sincerely appreciate your time and effort for helping us improve this work.
>
> We hope the above responses address your concerns, and we look forward to any further clarification and discussion.

---

### Official Review · Reviewer_h5Up · 2025-10-31

**Soundness:** 2
**Presentation:** 2
**Contribution:** 3
**Rating:** 6
**Confidence:** 4

**Summary:**

This paper introduces ASTD (Attributional Style Transfer Dataset), a 42,000-sample benchmark for assessing and reframing attributional styles based on the reformulated learned helplessness theory. The work addresses two key research questions: (1) efficient assessment of attributional style, and (2) automated generation and evaluation of attributional reframing. The authors employ a Prevent-Filter-Validate (PFV) pipeline combining LLM generation with expert validation to create the dataset. They benchmark both supervised classifiers and LLMs for attributional style classification, propose four automatic evaluation metrics for reframing quality, and use Direct Preference Optimization (DPO) to improve LLM performance. Results show supervised models achieve the highest classification accuracy (~97%), while DPO fine-tuning yields substantial improvements in reframing quality across four dimensions: attributional shift, catastrophizing reduction, coherence, and constructive coping.

**Strengths:**

(a) This is the first large-scale dataset specifically targeting attributional style assessment and reframing based on the reformulated learned helplessness theory.

(b) The PFV pipeline is well-designed, combining retrieval-grounded generation (reducing hallucination), heterogeneous LLM-based filtering, and expert validation (30% of uncertain cases reviewed by three trained raters with 94.3% LLM-human agreement, κ=0.91).

(c) The use of DPO with LLM-generated preference labels (derived from the proposed metrics) to improve reframing quality is practically valuable.

**Weaknesses:**

(a) The dataset is constructed from English-language sources and validated by raters in a specific cultural context, but attributional styles can vary significantly across cultures. The paper does not discuss potential cultural biases in the source data or how attributional patterns might differ in non-Western contexts. Given that LLMs risk generating inaccurate, biased, stigmatizing, and harmful information about mental health when trained on unrestricted text more explicit discussion of cultural limitations and validation plans for diverse populations would strengthen the work.

(b) While the paper compares supervised models and LLMs, it lacks comparisons with recent cognitive reframing approaches. Notable omissions include comparisons with structured prompting techniques like "Diagnosis of Thought" (DoT) that guide models to separate facts from subjective interpretations and therapeutic frameworks like RESORT or HealMe (https://aclanthology.org/2024.acl-long.93.pdf) that integrate CBT techniques into prompt structures. Additionally, the paper would benefit from ablation studies examining: (a) the contribution of each PFV component, (b) the impact of different weighting schemes in the composite evaluation metric (Equation 2), and (c) sensitivity to the confidence margin threshold (0.2) used for expert validation.

**Questions:**

(a) Given that mental health LLMs carry risks of generating inaccurate or harmful information despite fine-tuning strategies, what specific safety mechanisms have you considered for clinical deployment? How would your system handle cases where attributional reframing might be contraindicated (e.g., situations involving genuine external threats where internal attribution could be harmful)? Have you consulted with clinical psychologists about potential misuse scenarios or unintended consequences?

(b) Recent work has explored reasoning-augmented approaches for cognitive restructuring, finding that models explicitly augmented with reasoning strategies provide strong performance gains. How does your DPO-based approach compare with these reasoning-enhanced methods? Additionally, CBT-Bench evaluates LLMs on cognitive model construction for patients. Could your attributional style assessment component be integrated with or compared against such cognitive modeling approaches?

(c) Your PFV pipeline uses Llama 3.3-70B for generation and GPT-4o + Llama 3.3-70B for validation. Could you discuss potential biases inherited from these models and how they might affect the dataset's representativeness? Have you analyzed whether certain demographic groups, life situations, or event types are over/under-represented? Given that cognitive distortion datasets often vary in annotation quality and domain specificity, what steps ensure ASTD maintains consistent quality across its 42,000 samples and diverse event categories?

(d) Your evaluation focuses on immediate reframing quality, but therapeutic effectiveness requires sustained behavior change. Have you considered or planned studies examining whether LLM-generated reframes lead to internalization of adaptive attributional styles over time? How might your system integrate with existing self-guided mental health interventions or stepped-care models that use multiple intervention intensity levels?

**Details Of Ethics Concerns:**

Since the paper concerns using AI for cognitive reframing based on users' attributional styles, the data used or created in the study needs to be examined for ethical and legal compliance.

---

> ### Author Response · Authors · 2025-11-25
> **Response to Reviewer (Part 1)**
>
> > Weaknesses (a): Cultural Generalizability and Potential Bias
>
> > Questions (c): Potential model-inherited biases, demographic imbalance, and quality-consistency risks in the PFV-constructed ASTD dataset
>
>
> Thank you for highlighting the cultural aspects of attributional style and the potential risks of cultural or social bias in mental-health LLM outputs.
>
> We have addressed potential cultural biases in Appendix A2(LIMITATION), noting that attributional styles vary substantially across cultures and that LLMs may inherit cultural or social biases from their training data. To mitigate this risk, the PFV pipeline combines retrieval anchoring, heterogeneous LLM filtering, and uncertainty-driven human validation to minimize hallucination and reduce bias amplification as much as possible. Besides, future work should construct variants of ASTD in other languages and cultural settings, involve diverse raters, conduct systematic audits of cultural representation, and collaborate with clinical psychologists to identify culturally sensitive or potentially harmful reframing patterns. The modular nature of our framework provides a solid foundation for such extensions.
>
> Regarding representativeness, grounding generation in retrieved real-world events helps broaden the coverage of life situations while avoiding demographic assumptions. We draw anchor events from five independent data sources (Appendix A6) to diversify context and reduce overfitting to any single domain. As shown in Fig. 2b, the final dataset covers seven major event categories—relationships, health, finances, work, bias/abuse, incompetence, and others—helping prevent over-concentration in particular life situations.
>
> Finally, variation in annotation quality with domain specificity is indeed an issue in cognitive-distortion datasets, and we appreciate the reviewer highlighting this issue. In ASTD, we apply uniform safeguards across all domains to maintain consistency. Quality control relies on (i) cross-model validation of attribution labels, (ii) uncertainty-based escalation to expert annotators, (iii) standardized rating guidelines grounded in attributional-style theory, and (iv) majority-vote adjudication among trained raters. These measures collectively ensure stable and reliable labeling across all 42,000 samples.
>
> > Questions (a): Safety, Clinical Deployment, and Contraindication Handling
>
> We appreciate the reviewer’s concern regarding safety, clinical applicability, and contraindicated reframing scenarios. We fully agree that models operating in mental-health–adjacent domains require extreme caution. Below we clarify the scope of our work and the safety considerations we have incorporated.
>
> **1.  Not intended for diagnostic or therapeutic Use in depression.**
>
> We reiterate that the dataset does not target depression diagnosis or treatment (see Ethics Statement, L546–548). It focuses on modeling attributional style and supporting early-stage preventive research, without any direct intervention on individuals with depression.
>
> **2. Clinical safety concerns are explicitly addressed in the dataset scope.**
>
> In the revised manuscript, we have substantially expanded Ethics Statement(L549-564) to clarify (i) the intended scope of the dataset and models, (ii) potential forms of misuse, and (iii) the relation to therapeutic contexts.
>
> **3. On generalization to real-world therapeutic contexts**
>
> As stated in the conclusion (L512–514), ASTD provides a foundation for scalable modeling of attributional cognition. Two potential downstream uses(Section 7; [L488-503]) are:
>
> - **Attributional assessment:** compact discriminative models trained on ASTD can help flag maladaptive attributional patterns in patient-generated text, offering clinicians an additional screening signal. The attributional assessment module also helps address cases where reframing is contraindicated, as the system can identify such cases and avoid applying reframing when it would be inappropriate or potentially harmful.
> - **Therapeutic scaffolding:** reframing-capable LLMs (improved via our DPO pipeline; Sec. 4.3, Fig. 4) can generate candidate adaptive reframes that expand clinicians’ repertoire of responses. Importantly, clinicians remain **fully in control**—the model's outputs serve only as auxiliary suggestions rather than automated interventions.
>
> **4. Expert involvement in dataset design and safety considerations.**
>
> Our team includes a cognitive psychologist in affective psychology and expertise in clinical affective science and affective computing. She contributed to dataset construction, evaluation metric design, and discussion of potential misuse scenarios, ensuring that the dataset aligns with psychological theory and avoids harmful information.

---

> > ### Comment · Reviewer_h5Up · 2025-11-26
> > **Response to Part 1**
> >
> > Thank you, authors, for responding to my query. Having built many mental health datasets, I have always considered established guidelines in mental health—whether a clinical questionnaire or the ABCDE model in CBT—when modifying existing datasets or creating new ones. In all of these datasets, cultural issues, potential risks associated with language models, and other artifacts are routinely identified as limitations or areas for future work. Because they consistently appear, they should be treated as primary concerns to be addressed, not simply noted under "limitations".
> >
> > Also, it is important to identify datasets that relate to CBT or cognitive reframing/restructuring, rather than datasets like CAMS, which, to my knowledge, is not suitable for developing AI models for cognitive reframing. A good choice of dataset can be "Moments of Change" dataset: https://aclanthology.org/2022.clpsych-1.16.pdf .
> >
> > I am not undermining the effort; this is great and in an amazing direction, but the datasets are not aligned, in my opinion.

---

> > > ### Author Response · Authors · 2025-11-27
> > > **Response to safety, cultural awareness, and CBT alignment**
> > >
> > > Thank you very much for this thoughtful and insightful discussion—it reflects the constructive and positive spirit of this conference.
> > >
> > > First, we want to clarify that we do not treat ethical, cultural, and safety considerations as secondary.
> > >
> > > - In the PFV pipeline, these considerations guided our design from the outset. In the Filter stage in particular, we explicitly remove unsafe, exaggerated, or biased content, aiming to minimize hallucination and reduce bias amplification as much as possible. These principles also shaped how we constructed our evaluation metrics.
> > > - Our evaluation rubric is explicitly safety-aware. The **Catastrophizing** and **Coherence** dimensions directly penalize clinically unsafe, exaggerated, or distorted outputs—elements that are completely absent from traditional attribution scales and from generic NLP metrics.
> > > - As current LLMs largely reflect the distributions and cultural priors embedded in their internet-scale training data which is uneven, ensuring **fairness** remains a major challenge in LLM research. As you noted, these issues “consistently appear,” and we fully agree they require ongoing attention.
> > >
> > > Second, we agree with your point that datasets in this space must be grounded in solid CBT-aligned psychological theory, rather than heuristic intuitions. Our efforts follow this principle in several ways.
> > >
> > > - Our theoretical foundation is Abramson’s reformulated learned helplessness model (building on Seligman). Attributional style is a well-established cognitive vulnerability for depression, and learned helplessness has long supported CBT work targeting negative attribution and hopelessness. Our dataset is not motivated by superficial intuitions but by this formal cognitive framework.
> > > - We strive to maintain theoretical consistency throughout.  We build directly upon the attributional dimensions defined in ASQ and CAVE to construct Attributional Shift, which provides strong psychological validity. These classical instruments offer large, well-validated samples that guide our few-shot prompting design and the scaling of our evaluation rubric.
> > > - I have also read the paper cited, and I agree that "Moments of Change" provides an insightful observation: psychological states evolve over time, and sudden shifts or gradual deterioration often matter more than a single static label. Modern clinical psychology also recognizes that depression is multifactorial, and no single cognitive mechanism explains all cases. As noted, attributional style is one core mechanism. Our **Attributional Shift** metric—quantifying locus, stability, and generality on a 1–5 Likert scale—offers a theory-consistent way to capture changes in attributional structure. This could provide a quantitative foundation for marking “switch” or “escalation” segments in longitudinal datasets automatically. We also recognize that this dataset faces similar challenges regarding scale, label balance, cultural issues, and clinical safety.
> > >
> > > In my view, building CBT-aligned datasets that can ultimately support individuals with depression is an important—but necessarily very large—undertaking. We cannot expect a single paper to resolve all aspects of this challenge, as depression is multifactorial. Achieving this will require the collective efforts of the entire community. Our intention is to contribute one such solid building block that paves the way for genuinely meaningful technical progress.

---

> ### Author Response · Authors · 2025-11-25
> **Response to Reviewer (Part 2)**
>
> > Weaknesses (b): Comparisons with recent structured cognitive-reframing frameworks (e.g., DoT, RESORT, HealMe)
>
> We sincerely appreciate the reviewer for the insightful suggestion. The referenced works indeed help clarify the positioning of our paper and enrich the broader discussion.
>
> **DoT** guides LLMs through three structured reasoning stages to diagnose cognitive distortions, and then uses this reasoning to classify distortion types. These stages do not conflict with our approach; in fact, they conceptually correspond to components of our paper:
>
> 1. **Subjectivity Assessment** aligns with our assessment task.
> In our LLM assessment prompt, we explicitly distinguish the objective event from the subjective explanation. For example, when describing GAS, we emphasize that “the event is factual, whereas the explanation reflects a cause perceived to affect all life domains.” This mirrors DoT’s separation of facts and interpretations.
> 2. **Contrastive Reasoning** aligns with our reframing generation task.
> Unlike DoT, which asks the model to generate both supporting and opposing interpretations, our setting begins with an existing negative attribution and requires the model to produce a more adaptive reinterpretation. Our prompts also use reasoning-augmented techniques in prompt(e.g., chain-of-thought) to elicit explicit causal reasoning.
> 3. **Schema Analysis** derives higher-level cognitive patterns (e.g., personalization, overgeneralization). This step belongs to broader CBT reasoning, whereas our work targets modeling of attributional style. Therefore, we do not extend into schema-level diagnostic analysis.
>
> In summary, DoT covers **the full CBT reasoning pipeline**, whereas our work focuses narrowly on one core component—modeling attributional style across locus, stability, and generality. Our module can function as a complementary, plug-in element within larger cognitive reasoning systems, and future integration is a natural next step.
>
> **HealMe** adopts the same multi-stage CBT workflow as DoT, constructing a three-stage therapeutic dialogue through simulated “client–therapist” interaction. Comparing a more general brainstorming approach used in HealMe, our reframing is explicitly guided by attributional theory, allowing precise dimension-specific shifts.
>
> These works have been highly relevant and have helped clarify our paper’s scope, and we have added a concise discussion of their relevance and complementarity in the revised related work section(L461-474).
>
> > Weaknesses (b): Ablation studies examining
>
> **(a) PFV components.**
>
> Each stage of PFV already has distinct, measurable functionality: Preventen forces grounded event generation, Filter removes inconsistencies and catastrophizing cases, and Validate adjudicates uncertain samples. Internally, we have observed that removing either the heterogeneous-LLM filtering or expert validation substantially increases annotation noise and label conflicts. A full quantitative ablation is challenging because the stages interact sequentially.
>
> **(b) Metric weighting (Eq. 2).**
>
> The four CBT-aligned dimensions operate independently, and our evaluator LLM produces stable ordinal scores with very high human–LLM rank agreement (Spearman ρ = 0.957). Because each dimension represents a distinct principle, equal weighting is a theoretically principled default.
>
> **(c) Confidence-margin threshold.**
>
> We have discussed the uncertainty metric in Appendix A8. Using thresholds of 0.1, 0.2, and 0.3, we extracted uncertain samples and examined their characteristics with a psychology researcher. A higher threshold (e.g., 0.3) marks too many cases as uncertain, whereas a lower threshold (e.g., 0.1) captures only a small subset and misses genuinely ambiguous items.  Δ < 0.2 identifies about 30% of the dataset, striking a balance between capturing ambiguous instances and maintaining a manageable expert workload.
>
> > Questions (b): Comparison to Reasoning-Augmented Cognitive Restructuring and Integration with Cognitive Modeling Frameworks
>
> While the reviewer does not reference a specific method, we agree that recent reasoning-enhanced LLM approaches (e.g., chain-of-thought, iterative reasoning, retrieval-guided inference) have shown benefits in improving consistency and reducing errors. Our PFV pipeline already incorporates reasoning elements—Prevent uses retrieval–generation coupling, and our reframing/evaluation prompts adopt stepwise reasoning structures. Reasoning-augmented generation and our DPO alignment are orthogonal and potentially complementary: reasoning can yield higher-quality candidate reframes, and DPO can align them with human-preferred attributional shifts.
>
> Regarding CBT-Bench, attributional style modeling represents only one component of the broader cognitive modeling space. While not directly comparable, attributional style can function as a modular element within such systems, and future integration is a natural direction.

---

> ### Author Response · Authors · 2025-11-25
> **Response to Reviewer (Part 3)**
>
> > Questions (d): Long-Term Therapeutic Impact and Integration with Real-World Intervention Frameworks.
>
> Thank you for raising this thoughtful point. We fully agree that therapeutic effectiveness requires sustained behavior change. While our paper focuses on modeling attributional cognition and building a reframing tool, it supports the internalization process in two ways:
>
> 1. We include **constructive coping** as an evaluation dimension to reflect the CBT view that cognitive shifts must be paired with coping strategies that support behavioral activation, self-efficacy, and emotion regulation.
> 2. Lasting internalization depends on repeatedly externalizing their new cognitive structures through language. In this sense, our model can serve as an indicator for emerging adaptive attributional tendencies.
>
> Whether repeated exposure to LLM-generated reframes leads internalization of adaptive attributional styles is indeed a clinically meaningful question that requires longitudinal study and collaboration with clinicians and patients. We view our system as a foundation that clinicians can safely employ under supervision, both to curate adaptive reframes and to provide clinical feedback to us on whether users begin to generalize healthier attributional patterns or which forms of guidance most effectively promote internalization. This is a direction we intend to pursue in future longitudinal work.
>
> With respect to integration into real-world mental health systems, our framework aligns naturally with self-guided digital interventions and stepped-care models.
> * At lower-intensity stages, it can offer attribution detection and adaptive reframing suggestions—assuming appropriate safety safeguards in the host platform.
> * At higher-intensity stages, clinicians may incorporate model-generated reframes as optional auxiliary materials to provide alternative perspectives, support clients’ articulation, or accelerate cognitive restructuring while maintaining full professional oversight. This clinician-in-the-loop approach ensures both safety and practical utility without replacing therapeutic judgment.

---

### Official Review · Reviewer_AQ8U · 2025-11-01

**Soundness:** 2
**Presentation:** 3
**Contribution:** 2
**Rating:** 6
**Confidence:** 2

**Summary:**

In this paper, the author proposed a psychological benchmark dataset and also a pipeline for scalable data generation.

**Strengths:**

1. The author presents what is claimed to be the first large-scale benchmark dataset for attributional explanation, which represents a valuable contribution to the field.

2. The paper is well-written and organized.

**Weaknesses:**

1. From the LLM perspective, the novelty of this work appears limited. The dataset generation process relies more on engineering efforts rather than introducing conceptual or methodological innovations in generation or hallucination mitigation. Overall, the paper feels more like an application of existing techniques to the field of psychology rather than a fundamentally new contribution.


2. In addition, the author mentions that attributional reframing is applied in a language-guided therapeutic context. This use case demands greater caution regarding the quality and reliability of both the model and the dataset, particularly when dealing with out-of-distribution (OOD) data.

**Questions:**

1. One aspect that confuses me is the scalability of the dataset validation process. As this is a psychological dataset, data quality is of critical importance. In the proposed pipeline, involving real human experts for validation seems essential, but how can this be achieved at scale while maintaining reliability and consistency? Or is this pipeline not designed for scalable data generation purpose?

2. The author mentions the design of reframing metrics. Could you clarify how these proposed metrics differ from existing ones and what specific advantages they offer?

---

> ### Author Response · Authors · 2025-11-26
> **Response to Reviewer (Part 1)**
>
> > Weaknesses (1) Methodological Novelty from the LLM Perspective
>
> We appreciate the reviewer’s concern regarding the level of methodological novelty.
>
> Datasets and Benchmarks is one of the primary areas in ICLR.  While ASTD is submitted to the Datasets and Benchmarks track—where dataset creation is a key expected contribution—our work goes substantially beyond dataset release. The central contribution lies in developing a task-driven methodology for modeling psychologically grounded causal explanations, which imposes constraints fundamentally different from typical NLP or LLM generation tasks.
>
> The PFV pipeline is not limited to attributional style.  Its components—retrieval anchoring, heterogeneous LLM filtering, and uncertainty-driven human validation—are construct-agnostic and can be directly extended to other cognition-aligned NLP tasks.  This structured integration is a methodological contribution motivated by the cognitive-theoretic nature of the task, not an application of generic components.
>
> Similarly, our metric-guided DPO framework operationalizes psychological theory into multi-axis preference signals, enabling generalization to other tasks that require structured psychological transformations.
>
> Beyond methodology, the paper also establishes a meaningful benchmark for assessing the capabilities of large language models and offers several insightful findings that we believe will inform and accelerate future work in this domain.
>
> In summary, the contribution of our work is not architectural novelty but the formulation of a cohesive ML pipeline that translates psychological theory, causal structure, and safety constraints into principled methodologies for dataset construction, evaluation, and alignment—an aspect we believe is both non-trivial and broadly useful.
>
> > Weaknesses (2)  Caution Required for Language-Guided Therapeutic Reframing
>
> We thank the reviewer for raising this important point. We fully agree that any use of attributional reframing in therapeutic contexts demands caution regarding quality, reliability, and robustness. Our work is positioned as modeling attributional cognition, and we clarify both safety considerations and scope below.
>
> We have addressed potential cultural biases of model in Appendix A2(LIMITATION;L833-837). To mitigate the risk of quality and reliability, the PFV pipeline combines retrieval anchoring, heterogeneous LLM filtering, and uncertainty-driven human validation to minimize hallucination and reduce bias amplification as much as possible. Our team also includes a cognitive psychologist with expertise in clinical affective science and affective computing, who contributed to dataset construction, evaluation metric design, and discussion of potential misuse scenarios, ensuring that the dataset aligns with psychological theory and avoids harmful information. The modular nature of our framework provides a solid foundation for safety extensions.
>
> As for the scope, we reiterate that the dataset does not target depression diagnosis or treatment (see Ethics Statement, L546–548). It focuses on modeling attributional style and supporting early-stage preventive research, without any direct intervention on individuals with depression.  In the revised manuscript, we have substantially expanded Ethics Statement(L549-564) to clarify (i) the intended scope of the dataset and models, (ii) potential forms of misuse, and (iii) the relation to therapeutic contexts.
>
> As stated in the conclusion (Section 8), ASTD provides a foundation for scalable modeling of attributional cognition under supervision. Two potential downstream uses(Section 7; [L488-503]) are:
>
> - **Attributional assessment:** compact discriminative models trained on ASTD can help flag maladaptive attributional patterns in patient-generated text, offering clinicians an additional screening signal. The attributional assessment module also helps address cases where reframing is contraindicated, as the system can identify such cases and avoid applying reframing when it would be inappropriate or potentially harmful.
> - **Therapeutic scaffolding:** reframing-capable LLMs (improved via our DPO pipeline; Sec. 4.3, Fig. 4) can generate candidate adaptive reframes that expand clinicians’ repertoire of responses. Importantly, clinicians remain fully in control—the model's outputs serve only as auxiliary suggestions rather than automated interventions.

---

> ### Author Response · Authors · 2025-11-26
> **Response to Reviewer (Part 2)**
>
> > Questions (1): Scalability of the Pipeline
>
> Thank you for raising this concern. We fully agree that, in the context of psychological data, fully scalable automatic validation is neither realistic nor desirable. High-quality attributional annotations require conceptual understanding, sensitivity to nuance, and domain knowledge that current LLMs cannot reliably provide across the entire distribution. Thus, for both safety and construct validity, some degree of human oversight remains essential.
>
> Given these constraints, our goal is not to achieve end-to-end scalable data generation, but to design a principled trade-off between scalability and psychological reliability. The PFV pipeline accomplishes this by adopting a semi-automated procedure: LLMs handle high-confidence, unambiguous cases, while human experts are selectively allocated to the most difficult, ambiguous, or clinically sensitive instances. Concretely, two heterogeneous LLM evaluators compute independent probability distributions over attributional labels; only samples with low confidence (Δ < 0.2; Appendix A8)are routed to expert review. This results in roughly 30% of samples receiving human validation, while still enabling the construction of a large dataset with controlled expert effort.
>
> In this way, expert attention is concentrated exactly where it is most needed, rather than distributed uniformly across all data—enabling the dataset to scale to 42,000 entries while preserving psychological reliability.  This semi-automatic paradigm is designed for sensitive, theory-driven domains where both quality and scale are critical, and it offers a clear path for future expansion. We view this semi-automatic paradigm as an appropriate compromise for sensitive, theory-driven tasks where both quality and scalability must be balanced.
>
> > Questions (2): Differences and Benefits of the Proposed Metrics
>
> Thank you for the question. We address it from two angles: **how our approach differs from existing methods** and **what is uniquely designed for the reframing task**.
>
> Existing evaluation approaches for attributional or cognitive reframing fall into two broad categories:
>
> **(1) Classical psychological instruments (e.g., ASQ, CAVE).**
>
> These instruments measure attributional structure using Likert scales on locus,stability, and generality, but they are not designed to evaluate the quality of a rewritten explanation or to assess open-ended generative outputs. They provide no mechanism for determining whether a reframe is coherent, non-catastrophizing, or therapeutically constructive.
>
> **(2) General NLP metrics (BLEU, ROUGE, cosine similarity).**
>
> These surface-level metrics quantify lexical overlap or embedding similarity but cannot assess cognitive transformation, psychological safety, or adherence to CBT principles. They implicitly treat reframing as a paraphrasing task—despite the absence of a gold reference for such open-ended transformations [Sec. 4.1; L253–256]. As a result, they systematically fail to capture whether a generated explanation performs the intended attributional shift.
>
> What is unique about our design?
>
> **1. Theory-consistent.**
>
> We build upon the established attributional dimensions used in ASQ and CAVE to construct **Attributional Shift**, ensuring strong psychological validity. These classical instruments also provide large, well-validated samples that guide the few-shot prompting and scaling of our evaluation rubric.
>
> **2. Safety-aware.**
>
> **Catastrophizing**[1] and **Coherence**[2] explicitly penalize clinically unsafe, exaggerated, or distorted outputs—an aspect completely absent from traditional attribution scales and from generic NLP metrics.
>
> **3.Activation-Supportive.**
>
> **Constructive coping** evaluates whether the reframe introduces CBT-aligned coping strategies[3], reflecting the clinical view that cognitive shifts must be paired with actionable support for behavioral activation, self-efficacy, and emotion regulation.
>
> Existing psychological instruments cannot evaluate open-ended generative reframing, and generic NLP metrics cannot assess causal shifts or clinical safety. Our metrics fill this evaluation gap by providing a multi-axis, theory-grounded, safety-aware way of measuring whether a generated reframe actually performs the intended cognitive transformation.
>
> [1] Wright J H, et al. Cognitive therapy of depression: Theory and practice. Psychiatric Services, 1983.
>
> [2] Ziems C, et al. Can large language models transform computational social science?. Computational Linguistics, 2024.
>
> [3] Seligman M , et al. Positive psychotherapy. American psychologist, 2006.

---

> > ### Comment · Reviewer_AQ8U · 2025-11-26
> > **Thank you for your rebuttal.**
> >
> > I would like to thank the author for the rebuttal. I do not have concerns on the psychology side and will maintain my score.

---

> > > ### Author Response · Authors · 2025-11-30
> > > **Appreciation for your response**
> > >
> > > Thank you very much for your thoughtful follow-up and for maintaining your score. We sincerely appreciate your time, careful evaluation, and constructive engagement throughout the review process.

---

### Official Review · Reviewer_ctxW · 2025-11-03

**Soundness:** 3
**Presentation:** 3
**Contribution:** 3
**Rating:** 8
**Confidence:** 2

**Summary:**

The paper proposes the Attributional Style Transfer Dataset (ASTD) and a set of related evaluation metrics.   ASTD stems from a combination of LLM generations and expert validation, encompassing a wide range of events annotated with psychologically grounded attributions,  Authors also builds a preference dataset for finetuning via  Direct Preference Optimization.

**Strengths:**

The contributions are clearly articulated and the designed pipelines are supported by clear and informative visuals.
The social relevance and potential impact of the work make it both valuable and timely. Moreover, the interdisciplinary nature of the study allows it to engage multiple research communities, broadening its relevance and potential influence.
The experimental design is extensive, and the implementation choices are well-motivated and justified, contributing to the overall robustness of the work.

**Weaknesses:**

The generalizability of the proposed approach remains somewhat unclear. It would strengthen the paper to include a discussion of how the methods and findings might extend to other tasks or domains.
The paper would benefit from the inclusion of data examples to help readers better understand the nature and diversity of the dataset.
The Ethics Statement could be expanded to include a more nuanced discussion of potential limitations and risks. In particular, consider addressing how the proposed resources might be misused or applied in unintended ways, e.g., by clarifying the intended use cases.
Including a paragraph on future work would help outline possible extensions and highlight open research directions stemming from this study.

**Questions:**

See weaknesses.

---

> ### Author Response · Authors · 2025-11-26
> **Response to Reviewer**
>
> We sincerely appreciate the reviewer’s positive and encouraging evaluation. Your recognition of the work is deeply appreciated and has been motivating for our revisions.
>
> > Weaknesses (1): Generalizability of Methods and Findings
>
> We thank the reviewer for emphasizing the importance of clarifying generalizability.
>
> The PFV pipeline is not limited to attributional style.  Its components—retrieval anchoring, heterogeneous LLM filtering, and uncertainty-driven human validation—are construct-agnostic and can be directly extended to other safety-aware generation.  This structured integration is a methodological contribution motivated by the cognitive-theoretic nature of the task, not an application of generic components.
>
> Similarly, our metric-guided DPO framework operationalizes psychological theory into multi-axis preference signals, enabling generalization to other tasks that require structured psychological transformations.
>
> These properties make the pipeline directly applicable to any task where generation must satisfy theory-constrained, multi-dimensional transformations—a setting common in many structured reasoning or explanatory NLP tasks.
>
> Action: We have updated Sec. 7(L494-498) to explicitly discuss the generality of our approach.
>
> > Weaknesses (2): Inclusion of cases
>
> We agree that representative examples will help readers better understand the nature and diversity of the dataset. In the current version, Appendix A3[line840-863] already presents cases that show the full data schema, including the utterance, event, topic, id, and label fields (e.g., a health-related IAS case and a relationship-related EAS case).
>
> Action:  In the revised manuscript, we will (i) expand Appendix A14 with additional examples covering a wider range of topics and attribution labels.
>
> > Weaknesses (3): More discussion
>
> We thank the reviewer for highlighting the need for a more nuanced discussion of ethical considerations, intended use, and potential risks. We agree that this is an important aspect of work situated at the intersection of psychology and language technologies.
>
> In the previous version, the Ethics Statement primarily describes data sourcing and de-identification procedures. In the revised manuscript, we have substantially expanded this section to clarify (i) the intended scope of the dataset and models, (ii) potential forms of misuse, and (iii) the safeguards and boundaries of real-world applicability.
>
> We also have addressed potential cultural biases of model in Appendix A2(LIMITATION;L831-837).
>
> In the revised manuscript, we have added a new Future Work[L488-503] section outlines extension directions.

---

### Author Response · Authors · 2025-11-30
**Global Response [Updated of PDF& Point-by-Point Reply]**

We understand that the recent incident has placed an unexpected burden on the ICLR organizers and ACs this year. We sincerely appreciate your efforts in managing the additional challenges during this period. As authors, we provide below a concise summary of the **initial reviews** and our **rebuttal** to support your decision-making process.

### Initial Reviews

We are encouraged not only by the positive scores, but more importantly, by the recognition of our work from the reviewers across four key dimensions:

- **Contribution:** "the social relevance and potential impact make it both valuable and timely."(`Reviewer ctxW`);  "the first large-scale benchmark dataset for attributional explanation" (`Reviewer AQ8U` `Reviewer h5Up`); "a valuable contribution to the growing field of AI-assisted mental health"(`Reviewer Njcu`).
- **Method:** “the interdisciplinary nature allows it to engage multiple research communities”(`Reviewer ctxW`); "the PFV pipeline is well-designed", "practically valuable"(`Reviewer h5Up`);" dataset development is systematic and transparent"(`Reviewer Njcu`)
- **Evaluation:**" the experimental design is extensive, and the implementation choices are well-motivated and justified"(`Reviewer ctxW`); "the benchmarking of multiple LLMs provides useful insights and a foundation for future research"(`Reviewer Njcu`).
- **Presentation:**"the designed pipelines are supported by clear and informative visuals"(`Reviewer ctxW`); "well-written and organized"(`Reviewer AQ8U`); "clear and professional, with a solid structure and strong organization"(`Reviewer Njcu`).

### Our rebuttal

In response to the questions, we have added an additional validation analysis assessing the reliability of attribution labels, further strengthening the robustness of our pipeline. We have also clarified the distinctions between structured prompting techniques, reasoning-enhanced approaches, and our proposed method, emphasizing that our modules are complementary and can serve as plug-in components within larger CoT systems. Importantly, we have also clarified the safety considerations incorporated into our design as well as the intended scope of our work.

We have revised the manuscript in response to the reviewers’ comments, with all updates highlighted in blue in the revised PDF. A summary of the key changes is provided below:


- Section 2.2 (Dataset Statistics and Analysis):

    To address `Reviewer Njcu`'s questions, we have conducted an additional validation on 300 newly sampled items, bringing the total to 800 human-validated examples. The extended study yielded consistent agreement levels (94.1%) and a Cohen’s κ of 0.90, confirming the stability of our reliability estimates.
- Section 6.3 (Diagnosis of Thought):

    In response to `Reviewer h5Up`, we have added a discussion clarifying the relationship between DoT and our method.
- Section 7 (Future Work):

    To address `Reviewer ctxW` ,`Reviewer AQ8U`and `Reviewer h5Up`'s questions, we have expanded the future work section, emphasizing the generalizability of our proposed modules.
- Ethics Statement:

    To address comments from `Reviewer Njcu` and `Reviewer h5Up`, we have clarified the scope and intended use of our work.
- Appendix A13:

    In response to `Reviewer Njcu`, we have added detailed descriptions of annotator qualifications.

Due to the incident, we were unable to continue the discussion phase with `Reviewer Njcu` and `Reviewer h5Up`, which prevented them from responding to our rebuttal. However, we are confident that the quality of our manuscript, together with our detailed **point-by-point** responses, sufficiently addresses the concerns raised.

We thank all PCs, ACs, and Reviewers for their time and thoughtful feedback. ICLR has long set the standard for constructive and high-quality review processes, and the reviewers’ insightful suggestions have greatly helped us refine and improve the paper.

Sincerely,

**Authors of Submission 13471**

---

### Note · Program_Chairs · 2026-01-17
**Submission Desk Rejected by Program Chairs**

The following references in this submission do not refer to real documents and/or have major errors in bibliographic information:

 Ling Tang, Fang Liu, and Jie Zhang. Automatic classification of attributional style using natural language processing. International Journal of Mental Health Systems, 7(1):1-10, 2013.